# Rod genesis driven by *mafba* in an *nrl* knockout zebrafish model with altered photoreceptor composition and progressive retinal degeneration

Fei Liu[1,2,3☯], Yayun Qin[1,4☯], Yuwen Huang[1], Pan Gao[1], Jingzhen Li[1], Shanshan Yu[1], Danna Jia[1], Xiang Chen[1], Yuexia Lv[1], Jiayi Tu[1], Kui Sun[1], Yunqiao Han[1], James Reilly[5], Xinhua Shu[5], Qunwei Lu[1], Zhaohui Tang[1], Chengqi Xu[1]*, Daji Luo[2,3]*, Mugen Liu[1]*

**1** Key Laboratory of Molecular Biophysics of Ministry of Education, College of Life Science and Technology, Huazhong University of Science and Technology, Wuhan, P.R. China, **2** State Key Laboratory of Freshwater Ecology and Biotechnology, Institute of Hydrobiology, The Innovative Academy of Seed Design, Hubei Hongshan Laboratory, Chinese Academy of Sciences, Wuhan, P.R. China, **3** University of Chinese Academy of Sciences, Beijing, China, **4** Maternal and Child Health Hospital of Hubei Province, Tongji Medical College, Huazhong University of Science and Technology, Wuhan, P.R. China, **5** Department of Biological and Biomedical Sciences, Glasgow Caledonian University, Glasgow, United Kingdom

☯ These authors contributed equally to this work.
* cqxu@hust.edu.cn (CX); luodaji@ihb.ac.cn (DL); lium@hust.edu.cn (ML)

**Data Availability Statement:** The NGS data are available under the GEO accession numbers: GSE160138 (RNA-seq) and GSE160140 (scRNA-seq).

## Abstract

Neural retina leucine zipper (*NRL*) is an essential gene for the fate determination and differentiation of the precursor cells into rod photoreceptors in mammals. Mutations in *NRL* are associated with the autosomal recessive enhanced S-cone syndrome and autosomal dominant retinitis pigmentosa. However, the exact role of Nrl in regulating the development and maintenance of photoreceptors in the zebrafish (*Danio rerio*), a popular animal model used for retinal degeneration and regeneration studies, has not been fully determined. In this study, we generated an *nrl* knockout zebrafish model via the CRISPR-Cas9 technology and observed a surprising phenotype characterized by a reduced number, but not the total loss, of rods and over-growth of green cones. We discovered two waves of rod genesis, *nrl*-dependent and -independent at the embryonic and post-embryonic stages, respectively, in zebrafish by monitoring the rod development. Through bulk and single-cell RNA sequencing, we characterized the gene expression profiles of the whole retina and each retinal cell type from the wild type and *nrl* knockout zebrafish. The over-growth of green cones and misexpression of green-cone-specific genes in rods in *nrl* mutants suggested that there are rod/green-cone bipotent precursors, whose fate choice between rod versus green-cone is controlled by *nrl*. Besides, we identified the *mafba* gene as a novel regulator of the *nrl*-independent rod development, based on the cell-type-specific expression patterns and the retinal phenotype of *nrl*/*mafba* double-knockout zebrafish. Gene collinearity analysis revealed the evolutionary origin of *mafba* and suggested that the function of *mafba* in rod development is specific to modern fishes. Furthermore, the altered photoreceptor composition and abnormal gene expression in *nrl* mutants caused progressive retinal degeneration and

**Funding:** This work was supported by the National Key Research and Development Program of China [2018YFA0801000] and the National Natural Science Foundation of China [31601026, 81670890, 31871260 and 31801041]. The funders had no role in study design, data collection and analysis, decision to publish, or preparation of the manuscript.

**Competing interests:** The authors have declared that no competing interests exist.

subsequent regeneration. Accordingly, this study revealed a novel function of the *mafba* gene in rod development and established a working model for the developmental and regulatory mechanisms regarding the rod and green-cone photoreceptors in zebrafish.

## Author summary

Vision is mediated by two types of light-sensing cells named rod and cone photoreceptors in animal eyes. Abnormal generation, dysfunction or death of photoreceptor cells all cause irreversible vision problems. *NRL* is an essential gene for the generation and function of rod cells in mice and humans. Surprisingly, we found that in the zebrafish, a popular animal model for human diseases and therapeutic testing, there are two types of rod cells, and eliminating the function of *nrl* gene affects the rod cell formation at the embryonic stage but not at the juvenile and adult stages. The rod cell formation at the post-embryonic is driven by the *mafba* gene, which has not been reported to play a role in rod cells. In addition to the reduced number of rod cells, deletion of *nrl* also results in the emergence of rod/green-cone hybrid cells and an increased number of green cones. The ensuing cellular and molecular alterations collectively lead to retinal degeneration. These findings expand our understanding of photoreceptor development and maintenance and highlight the underlying conserved and species-specific regulatory mechanisms.

## Introduction

Vision is involved in many fundamental behaviors of animals such as navigation, foraging, predator avoidance, and mate choice [1]. In humans, inherited retinal diseases are a major cause of irreversible vision impairment and blindness, causing serious physical discomfort and mental problems in the patients, who also face a heavy economic burden [2,3]. There are two types of light-sensing cells named rod and cone photoreceptors in vertebrate eyes. Rods are highly sensitive to dim light and function under conditions of low light, whereas cones respond to bright light and mediate color vision [4,5]. Rods and cones are derived from the same progenitor cells, which are regulated by several extrinsic signals and intrinsic transcription factors, such as *CRX*, *NRL*, and *NR2E3* [6,7]. Exploring the fundamental mechanisms controlling the fate determination and differentiation of photoreceptors in diverse animal models is of great value for understanding the development and evolution of photoreceptors and the pathogenesis of retinal degenerative diseases.

 Neural retina leucine zipper (*NRL*) is a large MAF family transcription factor expressed in the retina and pineal gland [8,9], and has been identified as the master gene determining the cell fate between rod versus cone photoreceptors in mammals. Disruption of *Nrl* in mice results in the complete loss of rod photoreceptors, accompanied by the presence of a large excess of S-cone-like photoreceptors [10,11]. Similarly, loss-of-function mutations in *NRL* cause the autosomal recessive enhanced S-cone syndrome (ESCS), which is characterized by the absence of rod response and enhanced S-cone function, in humans [12–14]. Moreover, ectopic expression of *Nrl* transforms the fate of cone precursors to rods in mice [15], suggesting that *NRL* is essential and sufficient for the fate determination and differentiation of rod photoreceptors. Additionally, NRL activates the expression of many rod-specific genes (such as *RHO*, *GNAT1*, *PDE6A*, *REEP6*, and *MEF2C*) synergistically with CRX and NR2E3 [16–21],

and represses the cone-specific genes (such as *Thrb* and *Opn1sw* in mice) either directly or through activating the expression of *NR2E3* [22,23].

In recent years, the zebrafish has emerged as a useful animal model for retinal degeneration and regeneration studies [24–27]. Interestingly, increasing evidence suggests that the current working model of photoreceptor development, which is largely based on studies in mice and humans, does not perfectly match the observations in the zebrafish [28–31]. For example, a portion of rods are derived from S-cone precursors in mice, but this phenomenon is not observed in the zebrafish [29]. In addition, our recent study has shown that deletion of *nr2e3*, another key factor determining rod fate [32,33], eliminates rods completely but does not increase the number of UV or blue cones (corresponding to S-cones) in the zebrafish [30]. This observation disagrees with the retinal phenotype of *Nr2e3* or *Nrl* knockout mice [10,32]. Whether rods can be derived from the precursors of S cones or other types of cones in zebrafish, and if so, how this process is regulated remain unclear.

In this study, we generated an *nrl* knockout zebrafish model by using the CRISPR-Cas9 technology and systematically investigated the developmental processes of rod photoreceptors. Surprisingly, we found two waves of rod genesis in zebrafish, which could be distinguished as *nrl*-dependent at embryonic stage and *nrl*-independent at post-embryonic stage. In addition, we observed a continuous increase of green cones in adult *nrl* knockout zebrafish. The cellular and molecular mechanisms underlying this unexpected retinal phenotype were further investigated, and a modified model for the developmental processes and regulatory mechanisms regarding the rod and green-cone photoreceptors in zebrafish was established.

## Results

### Knockout of *nrl* leads to reduction but not complete loss of rods in zebrafish

The *nrl* knockout zebrafish was generated via CRISPR-Cas9 targeting the exon 2 of *nrl*, which encodes the conserved N-terminal minimal transactivation domain (Fig 1A). Through several rounds of crossing and mutation screening, we obtained a homozygous *nrl* mutant zebrafish line carrying a frameshift mutation (c.230_237del8, p.P77Hfs*4) (Fig 1B). The mutation was predicted to result in a severely truncated Nrl protein without the DNA binding and transactivation domains. The *nrl* mRNA levels were significantly up-regulated in this mutant zebrafish line (referred to as *nrl*-KO hereinafter) (Fig 1C), possibly due to a negative feedback mechanism [34]. No alternative transcript of *nrl* was detected in the retinas of wild type and *nrl*-KO zebrafish. We also sought to detect the protein levels of Nrl in WT and *nrl*-KO retinas. Unfortunately, we could not obtain a working antibody against the zebrafish Nrl.

Histological analysis was performed to examine the retinal morphology of the *nrl*-KO zebrafish. Surprisingly, the presence of rod nuclei (labeled as RN in S1B and S1E Fig) suggested that rods are not completely eliminated by *nrl* knockout in zebrafish. We also observed that the photoreceptor and outer-segment layers were thinned in the *nrl*-KO retinas (S1 Fig). The rosette-like anomalies commonly observed in the outer nuclear layer (ONL) of *Nrl* or *Nr2e3* knockout mice [35,36] were not observed in the *nrl*-KO zebrafish (S1A and S1D Fig).

Deletion of *Nrl* has been reported to eradicate rods in mice [10,11]. To assess whether the *nrl*-KO zebrafish has rods, immunofluorescence analysis with the anti-zebrafish Rho antibody was performed on retinal sections from WT and *nrl*-KO zebrafish. Indeed, we could detect the rod outer segments from 14 dpf (day post-fertilization) until 18 mpf (month post-fertilization) (Figs 1D and S2). The rod outer-segment layer was significantly thinner in the *nrl*-KO zebrafish than in the WT zebrafish (S3A Fig). Additionally, retinal whole-mount immunostaining revealed a diluted pattern of rod outer segments in the *nrl*-KO zebrafish, especially in the

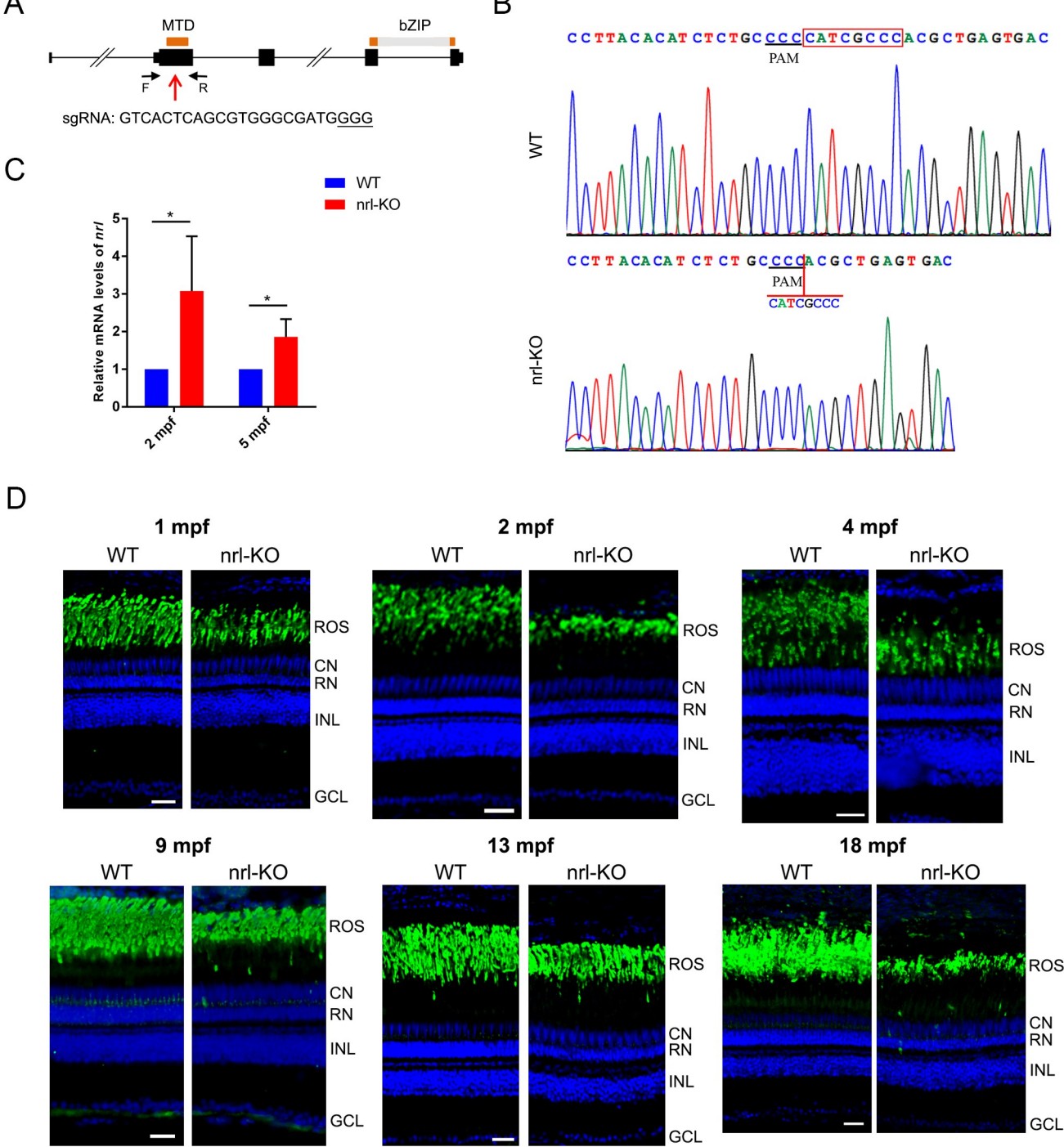

**Fig 1. Knocking out *nrl* diminishes but does not eradicate the rod population in zebrafish.** (A) The gene structure of zebrafish *nrl* and the CRISPR-Cas9 target site used for knocking out *nrl* are shown. Orange boxes, the exons encoding the MTD and bZIP domains; red arrow, the CRISPR-Cas9 target site; black arrows, the primers used for mutation detection. (B) Sequencing validation of the homozygous *nrl* del8 mutation (c.230_237del8). (C) The *nrl* mRNA levels in the *nrl*-KO zebrafish were measured using qPCR. The data are shown as mean with SD (n = 3). *, $p < 0.05$. (D) Immunostaining of retinal sections from WT and *nrl*-KO zebrafish from 1 mpf to 18 mpf with the anti-Rho antibody for the rod outer segments. Scale bars: 25 μm. ROS, rod outer segment; CN, cone nuclear layer; RN, rod nuclear layer; INL, inner nuclear layer; GCL: ganglion cell layer.

ventral retina (Figs 2 and S3B). Interestingly, we noticed that the rod generation in the marginal regions of *nrl*-KO retinas was delayed at 14 dpf and 1 mpf (S2A and S2B Fig). However, at 2 mpf and 4 mpf, these regions were covered by rods as in other retinal regions (S2C and S2D Fig), suggesting a central-to-peripheral rod development pattern in the *nrl*-KO zebrafish. Our results indicated that knockout of *nrl* in zebrafish reduced the number of rods rather than completely eliminating them, which was quite different from the retinal phenotypes of *Nrl* and *Nr2e3* knockout mice [10,37].

## Identification of two waves of rod genesis in the zebrafish

To trace the developmental processes of rods in the *nrl*-KO zebrafish, we used the Tg(rho: EGFP) transgenic zebrafish to label rods with EGFP (enhanced green fluorescent protein) *in vivo*. In WT zebrafish, rods first appeared in the retinal ventral edge and then spread to the entire retina with a rapidly increasing fluorescence intensity (Fig 3A and 3B). However, the fluorescence of rods in the *nrl*-KO zebrafish was undetectable or close to the background level until 7 dpf (Fig 3A). Interestingly, after 10 dpf, the fluorescence of rods in the *nrl*-KO zebrafish became obvious and rapidly increased with age (Fig 3B). Immunostaining the retinal sections for another rod-specific protein Gnat1 also showed the nearly complete absence of rods in the *nrl*-KO retinas at 7 dpf (Fig 3C). Flattened retinal whole-mounts of WT and *nrl*-KO zebrafish from 1 mpf to 3 mpf showed an increasingly extensive distribution of rods with age (Fig 4A), although the rod opsin expression level (reflected by the fluorescence intensity) was still lower in the *nrl*-KO retinas at 3 mpf than in age-matched WT controls (Fig 4A).

The protein levels of Rho (rhodopsin) were detected by dot blotting in WT and *nrl*-KO zebrafish (S4 Fig). Rho expression was indeed reduced in the *nrl*-KO retinas. Transmission electron microscopy was performed to determine whether the ultrastructure of the rod outer segments was affected in the *nrl*-KO zebrafish. There was no large difference in the length of the outer segments between *nrl*-KO and WT rods at 9 mpf (S5 Fig).

To discern the EGFP-labeled rods more clearly, retinal sections were prepared from the Tg (rho:EGFP) transgenic zebrafish at 1 mpf and observed at high magnification (Fig 4B). The *nrl*-KO groups had fewer rods than the WT controls, as suggested by the thinner rod nuclear layer (Fig 4B). Besides, the regions nearby ciliary marginal zone (labeled with boxes in Fig 4B) in the *nrl*-KO retinas showed no or little fluorescence signal of rods, consistent with our immunostaining results shown above (S2B Fig). Taken together, these results suggested that there are two waves of rod genesis in zebrafish. Knockout of *nrl* eliminates the first wave of rod genesis, starting at about 3 dpf (embryonic stage), but does not affect the second wave of rod genesis, starting at 7–10 dpf (juvenile stage). Accordingly, we named them as the *nrl*-dependent and *nrl*-independent rods, respectively.

## Down-regulation of rod-specific genes and increased number of green cones in adult *nrl*-KO zebrafish

To investigate the transcriptome dynamics in the *nrl*-KO retinas, we performed RNA-sequencing (RNA-seq) analysis on 2 mpf WT and *nrl*-KO retinas. A total of 386 differentially expressed genes (fold change $\geq 2$, adjusted *p*-value $\leq 0.05$, and FPKM $\geq 1$) were detected (Fig 5A and S1 Dataset). The functions and related biological processes of the top 100 differentially expressed genes were classified into several categories (S2 Dataset). Interestingly, we found that the most affected functional categories were inflammatory and immune response, signal transduction, metabolic enzymes, cell adhesion and extracellular matrix, transport, and cell protection and death (Fig 5B). These altered biological functions were similar to those identified in the *Nrl* knockout mice [38].

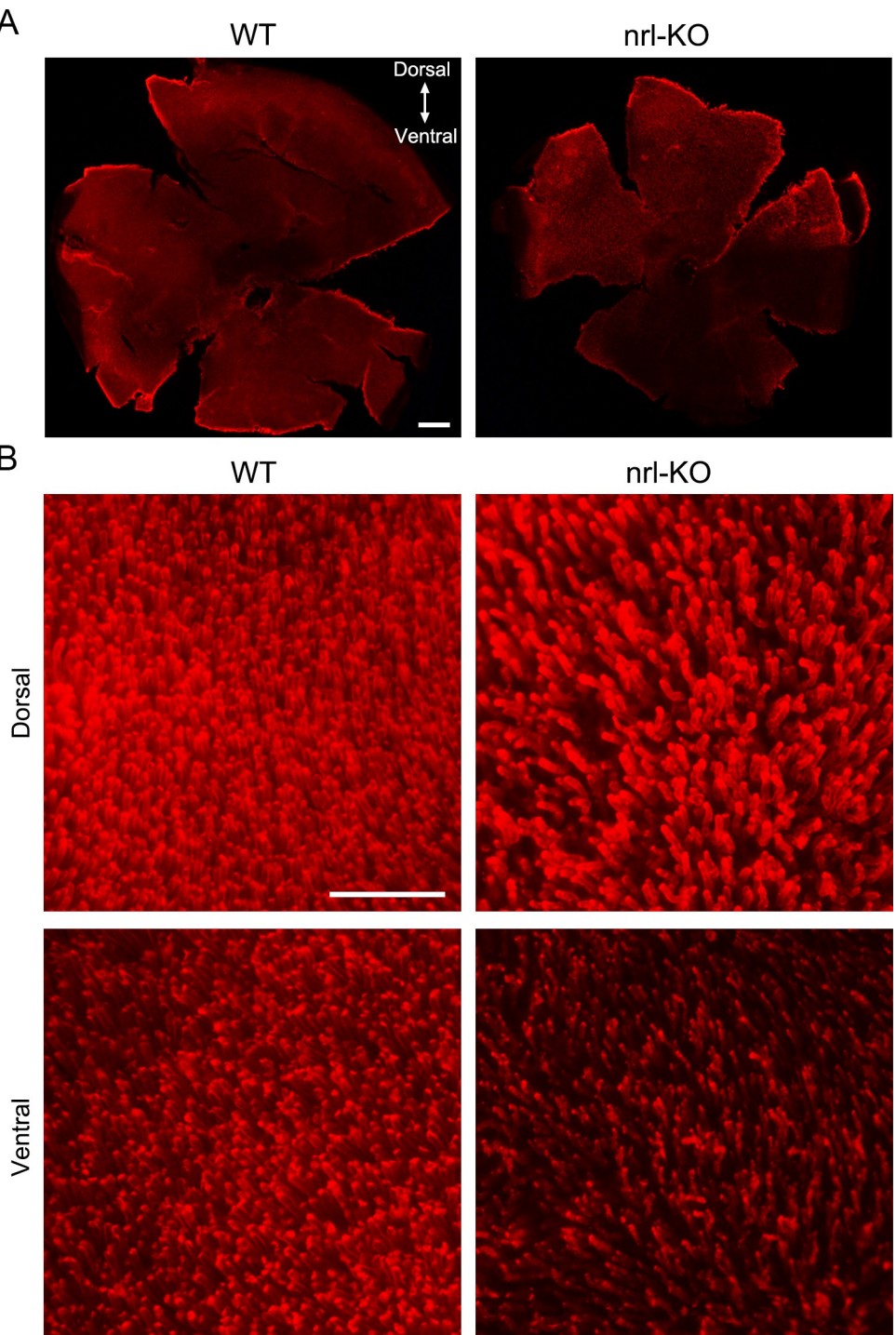

**Fig 2. Reduced density of the rod outer segments in the *nrl*-KO zebrafish.** (A) The rod outer segments were visualized via immunostaining on the flattened whole-mount retinas from WT and *nrl*-KO zebrafish at 2 mpf using the anti-Rho antibody. The overall views of WT and *nrl*-KO retinas are shown in the low-magnification images. Scale bar: 200 μm. (B) High-magnification images from the dorsal and ventral retinal regions are shown. The density of rods is reduced in the *nrl* knockout retinas, especially in the ventral retinal region. Scale bar: 50 μm.

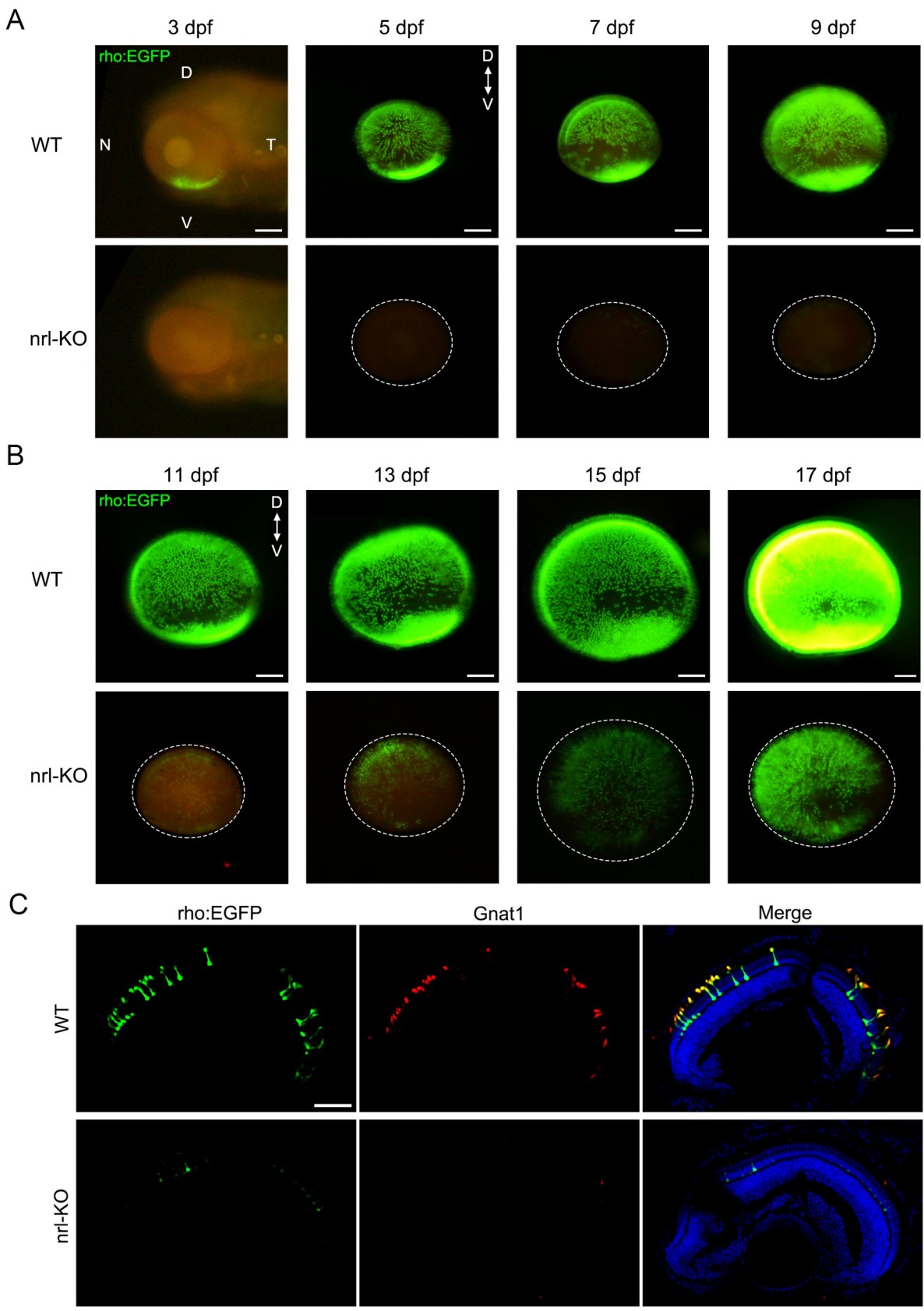

**Fig 3. Tracing the developmental processes of rods in WT and *nrl*-KO zebrafish.** Rods were labeled with EGFP by crossing the WT or *nrl*-KO zebrafish with the Tg(rho:EGFP) transgenic line. Fluorescence was observed every 2 days from 3 dpf. Representative images of WT and *nrl*-KO retinas are shown in (A) for 3–9 dpf and (B) for 11–17 dpf. The dotted circles indicate the boundaries of the retinas. D, dorsal; V, ventral; N, nasal; T, temporal. Scale bars: 100 μm. (C) Immunostaining of retinal sections for Gnat1 (the rod transducin alpha-subunit) showed exact co-localization with EGFP in rods at 7 dpf. Rods were barely observed in the *nrl*-KO retinas. Scale bar: 50 μm.

To our surprise, only few differentially expressed genes (*aipl1*, *nxnl1*, *sagb*, *gucy2f*, *rpgra*, and *cnga3b*) were closely related to the development or degeneration of retinas. Because we observed a reduced number of rods in the *nrl*-KO zebrafish, we directly examined the expression of rod- and cone-specific phototransduction genes by using our RNA-seq data. As expected, most of the rod-specific genes were down-regulated (adjusted *p*-value ≤0.05) in the *nrl*-KO group (Fig 5B). However, the fold changes were mostly less than 2, which were ignored by the routine bio-informatic pipelines. Meanwhile, the mRNA levels of cone-specific genes were not significantly changed, except for the obviously up-regulated green-cone opsins (Fig 5C).

To confirm the RNA-seq results, we measured the protein levels of representative rod-specific (*rho*, *gnat1*, and *gnb1*) and cone-specific (*gnat2* and *gnb3*) genes from 14 dpf to 3 mpf via western blotting. The rod-specific genes were significantly down-regulated in the *nrl*-KO retinas (Fig 6A and 6B). In contrast, protein levels of the cone-specific genes were unchanged before 2 mpf, and significantly increased at 3 mpf (Fig 6A and 6B). We also performed qPCR to measure the mRNA levels of cone opsin genes at 2 mpf and 5 mpf. The green-cone opsins (*opn1mw1*, *opn1mw2*, *opn1mw3*, and *opn1mw4*) were significantly up-regulated at both time points, whereas the expression of UV-, blue-, and red-cone opsins were almost unchanged (Fig 6C). These results correlated well with our RNA-seq data and suggested that the down-regulation of rod-specific genes and up-regulation of cone-specific genes (more particularly green-cone genes) both occur in the *nrl*-KO zebrafish.

To validate whether the number of green-cones is increased, we performed immunostaining using the anti-Opn1mw (zebrafish green-cone opsin) antibody on retinal sections from WT and *nrl*-KO zebrafish. No obvious changes were observed in the *nrl*-KO retinas at 2 mpf and 3 mpf, except for some newly formed green-cone outer segments located below the normal single-row arrangement (Fig 6D). However, at 13 mpf, 2–3 rows of green-cone outer segments could be observed (Fig 6D), suggesting a progressive increase in the number of green-cones. The expression levels of genes involved in cone development (such as *rx1*, *tbx2a*, *tbx2b*, *six7*, and *thrb*) were not significantly affected as suggested by the RNA-seq data.

## Single-cell RNA-seq revealed the cell compositions and rod/green-cone intermediate photoreceptors in *nrl* knockout retinas

The retinal cell compositions and cell-type-specific gene expression patterns in WT and *nrl*-KO zebrafish at 5 mpf were characterized via single-cell RNA-seq (scRNA-seq) analysis. After filtering out invalid cells, 10374 cells (4682 from the WT group and 5692 from the *nrl*-KO group) were classified into 24 clusters via unsupervised cell clustering analysis (Fig 7A). The relative mRNA levels of all the genes expressed in the 24 cell types were determined (S3 Dataset). According to the expression patterns of known marker genes of each retinal cell type [39–42], the 24 cell clusters were identified as rods (3 subtypes), cones (2 subtypes), horizontal cells, bipolar cells (12 subtypes), amacrine cells, retinal ganglion cells (RGCs), retinal pigment epithelium (RPE) cells, Müller glia, astrocytes, and microglia (S5 Fig and S4 Dataset).

The percentages of all the cell types in WT and *nrl*-KO retinas were analyzed (Fig 7B). In *nrl*-KO zebrafish, the percentage of rods was reduced by approximately 50%, whereas cones (more accurately the red/green cones) showed an obvious increase (Fig 7B and 7C). Three rod

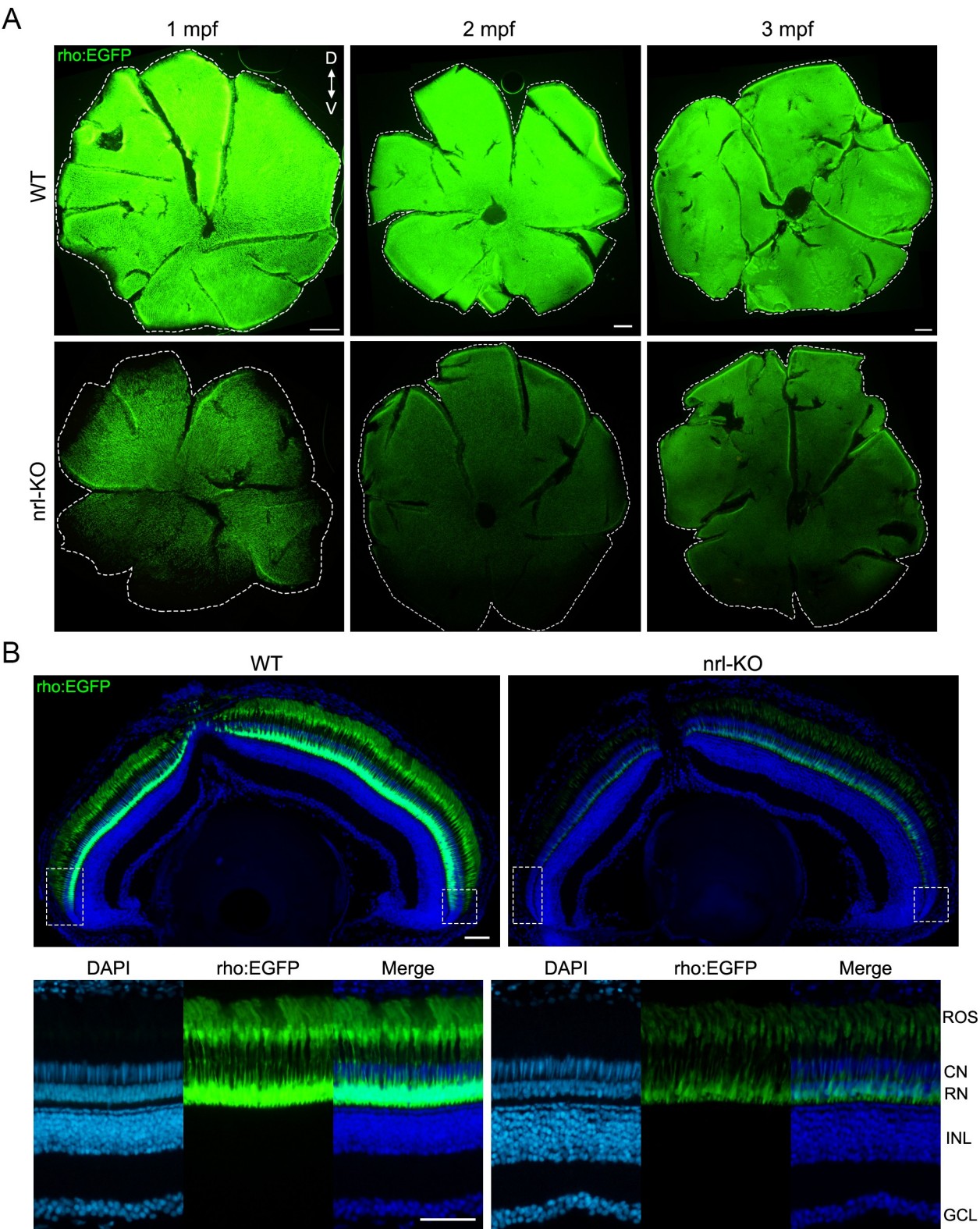

**Fig 4. Distribution of the rods and expression of the rod opsin in the WT and *nrl*-KO retinas from 1 mpf to 3 mpf.** (A) The Tg(rho:EGFP) transgenic line was used to label rods with EGFP. Representative images of flattened whole-mount retinas from WT and *nrl*-KO zebrafish at 1 mpf, 2 mpf, and 3 mpf are shown. The dashed lines indicate the edges of the retinas. Scale bars: 200 μm. (B) EGFP-labeled rods were observed on the retinal sections from WT and *nrl*-KO Tg(rho:EGFP) transgenic zebrafish at 1 mpf. The overall views are shown in the upper panel. The regions

nearby the ciliary marginal zone (labeled with boxes) showed no or little fluorescence signal of rods. Enlarged images of the dorsal retinal regions are shown in the lower panel. ROS, rod outer segment; CN, cone nuclear layer; RN, rod nuclear layer; INL, inner nuclear layer; GCL: ganglion cell layer. Scale bars: 50 μm.

subtypes (rod-1, rod-5, and rod-11) were identified in our scRNA-seq analysis. The top 20 marker genes of these subtypes are shown in S1 Table. The rod-1 population (accounting for 80% of all the rods) was the major subtype of rods in WT zebrafish, but the percentage was reduced to 13% in the *nrl*-KO zebrafish. Conversely, the percentages of rod-5 and rod-11 populations were increased to 34% and 16% of all the rods in the *nrl*-KO zebrafish, respectively. These data suggested that knockout of *nrl* affects both the fate determination between rods versus green cones and the ratios of rod subtypes.

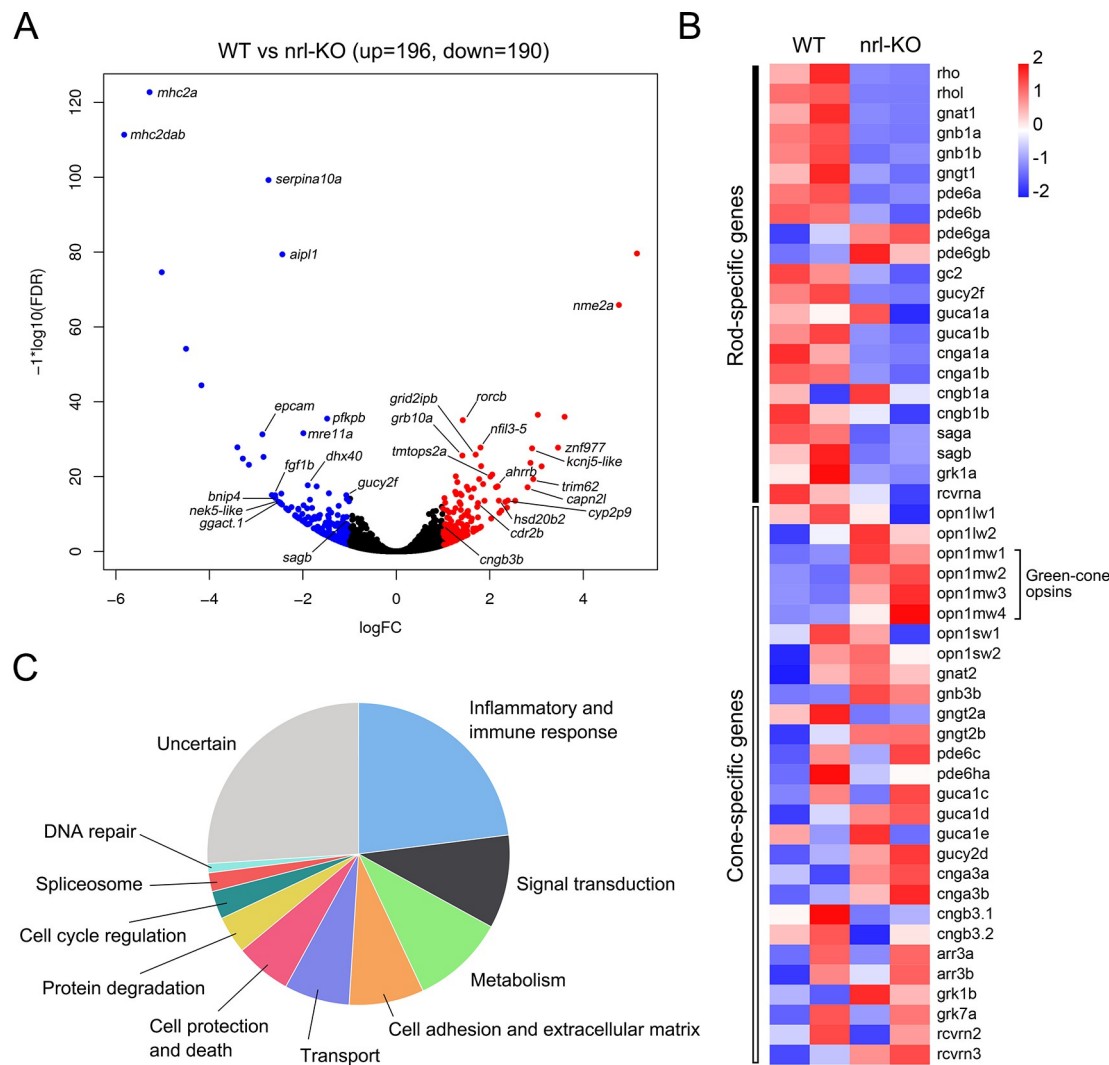

**Fig 5. Differentially expressed genes between WT and *nrl*-KO retinas.** (A) The volcano plot shows the 386 differentially expressed genes (196 up-regulated and 190 down-regulated) between WT and *nrl*-KO retinas at 2 mpf identified via RNA-seq. The red and blue points indicate the up-regulated and down-regulated genes, respectively. (B) The functional categories enriched among the top 100 differentially expressed genes. (C) The expression patterns of rod- and cone-specific phototransduction genes in *nrl*-KO retinas shown as a heatmap. Most of the rod genes were down-regulated in the *nrl*-KO retinas.

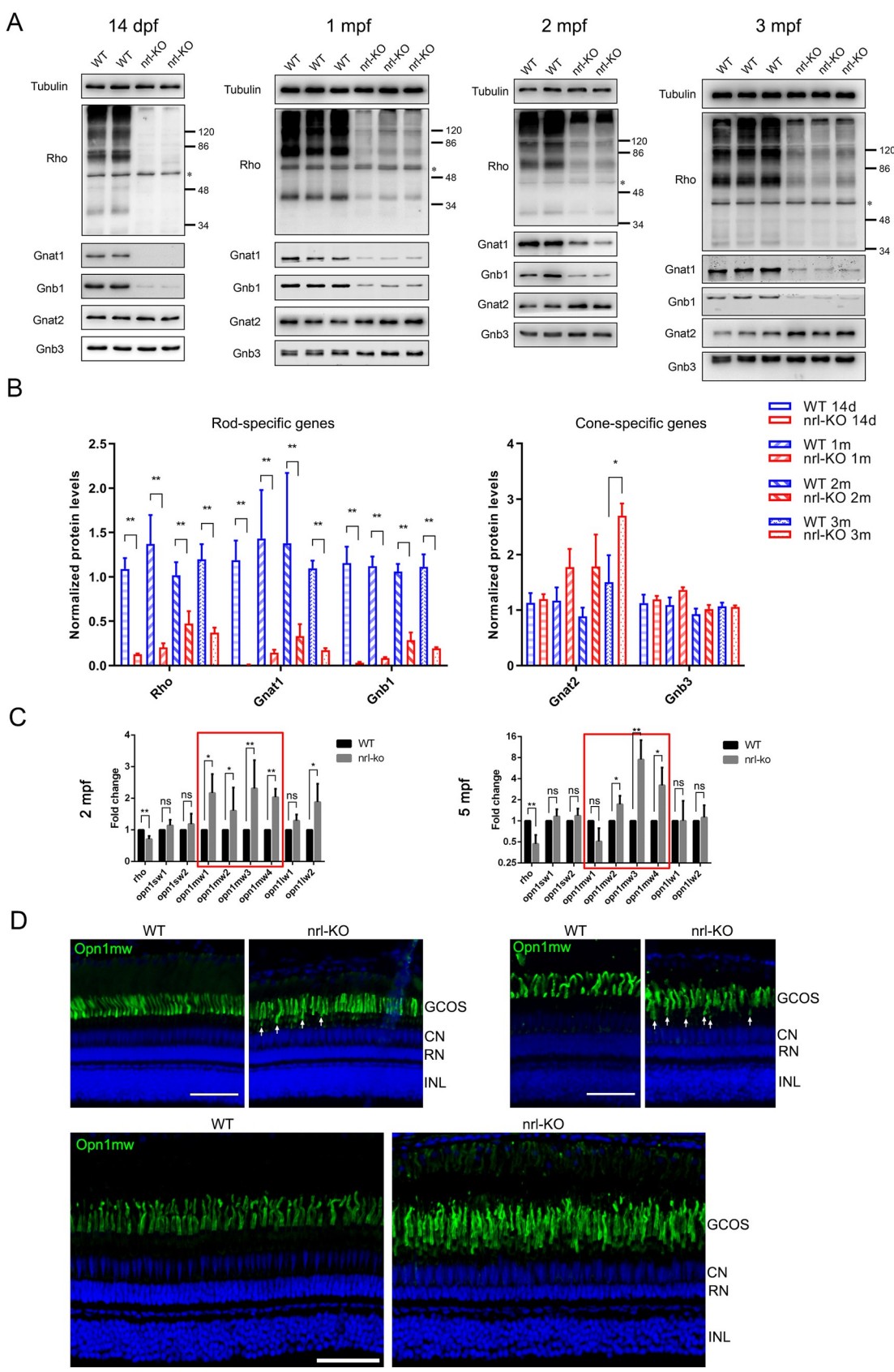

**Fig 6. Down-regulation of rod-specific genes and increased number of green cones in adult *nrl*-KO zebrafish.** (A) The protein levels of rod-specific (*rho*, *gnat1*, and *gnb1*) and cone-specific (*gnat2* and *gnb3*) genes in WT and *nrl*-KO retinas from 14 dpf to 3 mpf were evaluated using western blotting. Tubulin was used as a loading control. The asterisk indicates a non-specific band. (B) Quantitative analysis of the protein levels of rod- and cone-specific genes based on at least three independent experiments. WT samples, blue color; *nrl*-KO samples, red color. Data from different ages are arranged from left to right (from 14 dpf to 3 mpf) and indicated by different fill patterns. The data are shown as mean with SD (n = 3). *, $p < 0.05$. **, $p < 0.01$. (C) The mRNA levels of the rod and cone opsins in the WT and *nrl*-KO retinas at 2 mpf and 5 mpf were measured using qPCR. The data are shown as mean with SD (n = 3). ns, non-significant; *, $p < 0.05$; **, $p < 0.01$. (D) Detection of green-cones on retinal sections of WT and *nrl*-KO zebrafish at 2 mpf, 3 mpf, and 13 mpf by immunostaining using the anti-Opn1mw antibody. The dorsal retinal regions are shown. White arrows indicate the mislocalized green-cone outer segments. GCOS, green-cone outer segments; CN, cone nuclear layer; RN, rod nuclear layer; INL, inner nuclear layer. Scale bars: 50 μm.

The gene expression signatures of the three rod subtypes were characterized based on the gene expression profiles through unsupervised clustering analysis. The top 40 discriminative genes between the WT and *nrl*-KO rods are shown in a clustering heatmap (Fig 7D). The cells with high expression of rod-specific genes (such as *rho*, *rhol*, *saga*, *sagb*, and *gnat1*) and very-low expression of cone-specific genes (such as *opn1mw1*, *opn1mw3*, *arr3a*, *grk7a*, and *gucy2d*) represent the "typical" rods (leftmost in Fig 7D) in WT zebrafish. By contrast, the cells with relatively low expression of rod-specific genes and high expression of cone-specific genes (especially the green-cone opsins) most likely represent the "hybrid" rods (rightmost in Fig 7D) in the *nrl*-KO zebrafish.

To directly visualize these intermediate photoreceptors, immunostaining was performed on retinal sections from the Tg(rho:EGFP) transgenic zebrafish at 3 mpf with the anti-Opn1mw (green-cone opsin) antibody. Mis-expression and accumulation of Opn1mw in some rods (labeled by EGFP) were observed only in the *nrl*-KO retinas (Fig 7E). These results demonstrated the existence of rod/green-cone intermediate photoreceptors in the *nrl*-KO zebrafish.

Previous studies have shown the abnormal chromatin morphology of the S-cone-like cells in *Nrl* or *Nr2e3* knockout mice [10,32]. Interestingly, our scRNA-seq data showed that the *hmgn2* gene, which encodes a non-histone nucleosome binding protein and contributes to chromatin plasticity and epigenetic regulation [43], was remarkably up-regulated in *nrl*-KO rods. Through *in situ* hybridization, we observed that *hmgn2* was expressed at a very low level in WT rods (S6 Fig). However, in the *nrl*-KO zebrafish, the expression of *hmgn2* was specifically enhanced in the rods (S7 Fig). These results suggested that epigenetic modification is also involved in the formation of the intermediate photoreceptors in *nrl*-KO zebrafish.

## Identification of the *mafba* gene as a novel driving factor for rod development

A reasonable explanation for the occurrence of *nrl*-independent rods is that other genes might at least partly substitute for the functions of *nrl* in driving rod genesis. Genes belonging to the large MAF family (*MAFA*, *MAFB*, and *MAF/c-MAF*) are the most likely candidates due to their high homologies with NRL [29,44]. In addition, *MAFA* has been reported to induce the expression of rod-specific genes when ectopically expressed in mice [29,45]. By analyzing our scRNA-seq data, the expression patterns of all large MAF genes and genes involved in photoreceptor development were determined. We found that *mafba*, one of the two *MAFB* orthologs in zebrafish, showed a closer cell-type-specific expression pattern to that of *nrl* and *nr2e3* than the other MAF candidates (see the first three rows in Fig 8A). Furthermore, the *mafba* gene (indicated by green bars in S8A Fig) was the only large MAF family gene expressed in all the three types of rods at levels comparable to those of *nrl* and *nr2e3*.

To validate whether *mafba* plays a role in rod development, we knocked out this gene in zebrafish by using the CRISPR-Cas9 technology (S8B and S8C Fig), and then generated an *nrl*

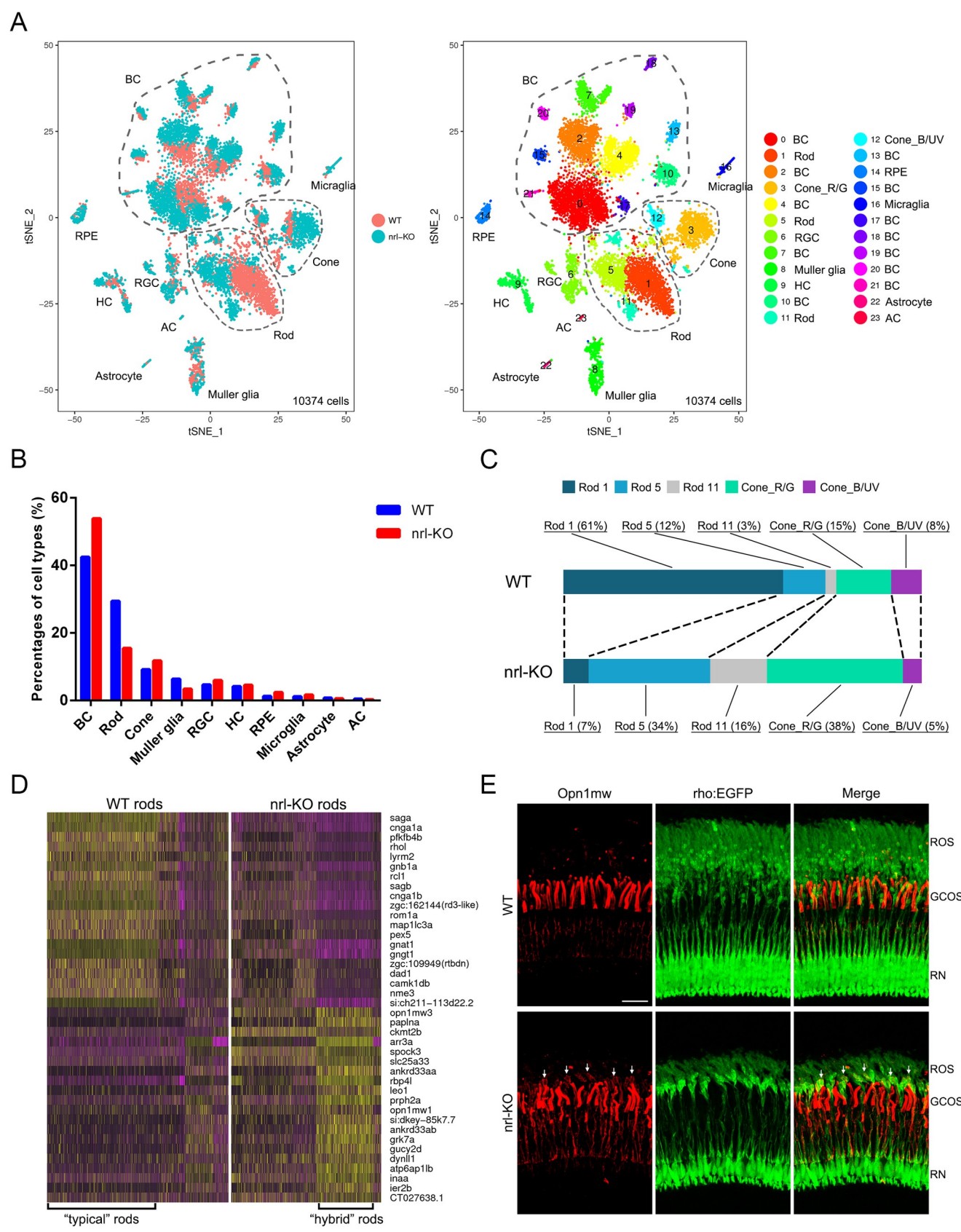

**Fig 7. Single-cell RNA-seq analysis in WT and *nrl*-KO retinas. (A)** tSNE visualization of the unsupervised cell clusters from 5-month-old WT and *nrl*-KO zebrafish. Left, the distribution of cell clusters between the WT and *nrl*-KO groups. Right, the retinal cell types identified via scRNA-seq (see also S6 Fig). BC, bipolar cells; Cone_R/G, red and green cones; Cone_B/UV, UV and blue cones; RGC, retinal ganglion cells; HC, horizontal cells; AC, amacrine cells; RPE, retinal pigment epithelium. **(B)** Proportions of each retinal cell type in the WT and *nrl*-KO zebrafish. **(C)** The cell compositions and proportions of photoreceptor sub-clusters in WT and *nrl*-KO retinas. **(D)** The heatmap shows the clustering pattern of the top 20 down-regulated and top 20 up-regulated genes in the WT and *nrl*-KO rods. Yellow, high expression; purple, low expression. **(E)** Mis-expression of green-cone opsin in a proportion of rods in the *nrl*-KO retinas. The dorsal retinal regions of WT and *nrl*-KO zebrafish at 3 mpf are shown. Green, EGFP-labeled rods. Red, immunofluorescence signals of the anti-Opn1mw antibody. The green fluorescent signal in the *nrl*-KO group was artificially enhanced to discern the morphology of rods. ROS, outer segments of rods; GCOS, outer segments of green cones; RN, rod nuclear layer. Scale bar: 20 μm.

and *mafba* double-knockout line by crossing the two single-knockout lines. The homozygous *mafba* mutants died at 9–10 dpf with multi-organ developmental defects, as described previously [46,47]. In the *nrl*-KO zebrafish, deleting both copies of *mafba* completely eliminated rods, based on the results of fluorescence analysis at 9 dpf (Fig 8B). Interestingly, deleting one copy of *mafba* also dramatically retarded the development of *nrl*-independent rods at 9 and 15 dpf, suggesting a dose-dependent effect of *mafba* (Fig 8B). We also measured the mRNA levels of *nrl*, *nr2e3*, and *mafba* through RNA-seq and qPCR. The expression of *mafba* in the *nrl*-KO retinas was significantly decreased at 2 mpf and 5 mpf (Fig 8C), excluding the genetic compensation effect caused by *nrl* disruption. Taken together, our results demonstrated that *mafba* is a novel regulatory gene driving rod genesis in the zebrafish.

To assess whether the function of *mafba* in rod development also exists in other species, the evolutionary origin of *mafba* was investigated via gene collinearity analysis. We found that the *mafba* and *mafbb* genes presumably arose through the duplication of an ancestral *mafb* gene specifically in the teleost fish ancestor (Fig 9). This genetic feature is not seen in ancient fishes, amphibians, reptilians, birds, or mammals. From European eel to zebrafish, there seems to be a chromosomal rearrangement upstream of the *mafbb* gene. The biological roles of the two genes may have gradually diverged during evolution, whereby *mafba* may have acquired the unique role in the rod development in modern fishes such as zebrafish and medaka.

## Progressive retinal degeneration and regeneration in the *nrl* knockout zebrafish

The gradual loss of rods is considered a main cause of the secondary death of cones in retinitis pigmentosa patients [48,49]. As the *nrl*-KO zebrafish showed a significantly reduced number of rods and over-growth of green cones, we assessed whether these developmental abnormalities lead to retinal degeneration. TUNEL assay was performed to evaluate cell death on retinal sections from 5 mpf to 13 mpf. Indeed, we observed more apoptotic cells in the *nrl*-KO retinas, including rods (predominantly), cones, RPE cells, and inner retinal cells (Fig 10A and 10B). Immunostaining for ZO-1 showed the abnormal morphology of some RPE cells in the *nrl*-KO retinas (Fig 10C). We also detected the up-regulation of GFAP, a hallmark of retinal damage, in aged *nrl*-KO retinas (Fig 10D). These results demonstrated the *nrl*-KO zebrafish undergo a slowly progressing retinal degeneration.

Progressive photoreceptor degeneration has been reported to activate regeneration in zebrafish [50]. Thus, we speculated that in the *nrl*-KO zebrafish, retinal regeneration might also be activated to replenish the lost rods. Immunostaining for Pcna on retinal sections indeed showed more proliferating cells in the *nrl*-KO retinas than in the WT controls since 9 mpf (Fig 10E and 10F). These results suggested that the progressive retinal degeneration in the *nrl*-KO zebrafish induces regeneration of photoreceptors in an *nrl*-independent manner.

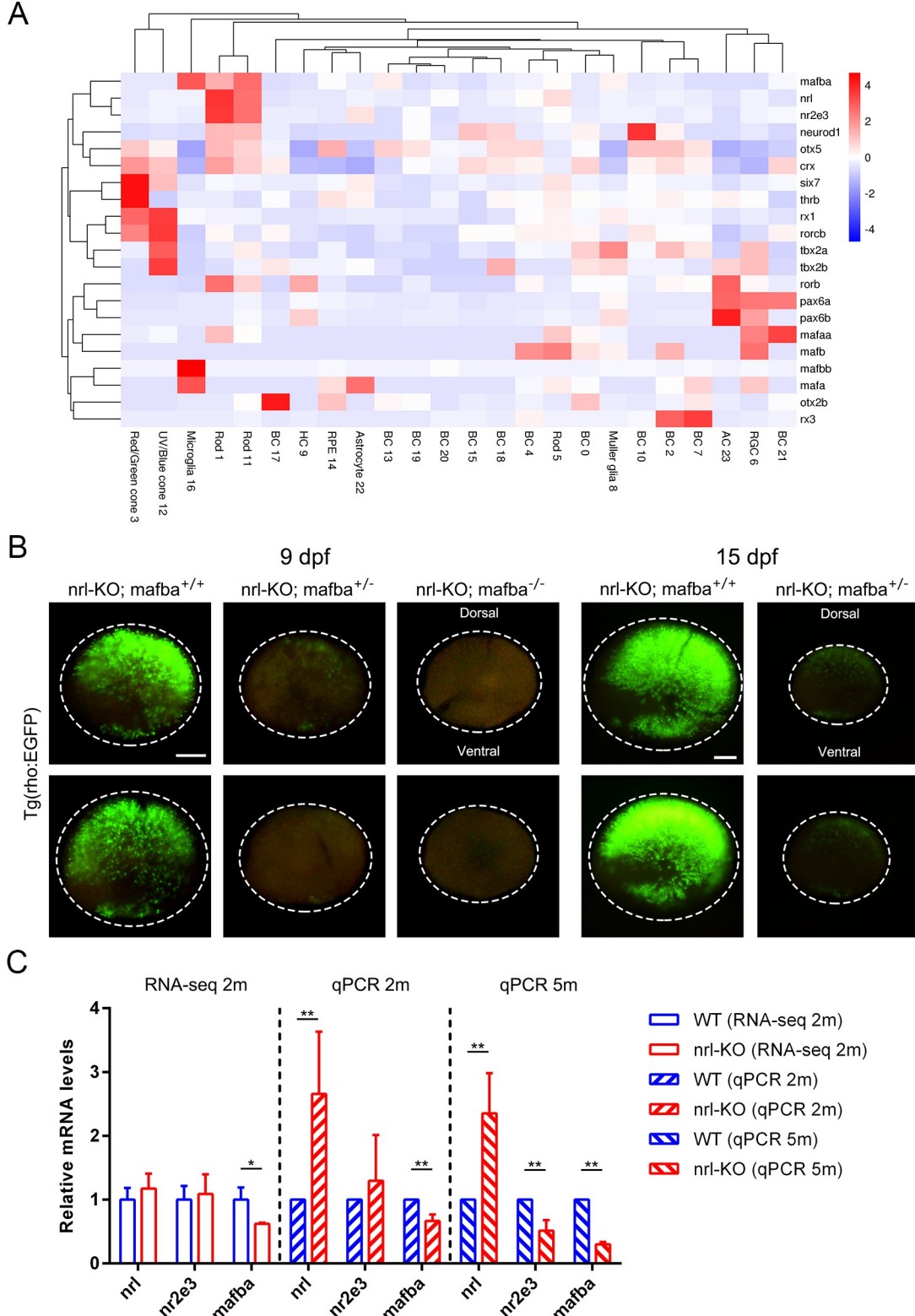

**Fig 8. Identification of the *mafba* gene as a novel driving factor for the development of *nrl*-independent rods.** (A) Clustering analysis of the large MAF genes and the genes involved in photoreceptor development by using the expression data from scRNA-seq. The cell-type-specific expression pattern of *mafba* was highly similar to those of *nrl* and *nr2e3*. (B) The distribution of rods (labeled with EGFP) in the *nrl*-KO zebrafish carrying the WT, heterozygous, and homozygous *mafba* alleles

at 9 dpf (left panel) and 15 dpf (right panel). Knocking out *mafba* further reduced the genesis of rods in the *nrl*-KO zebrafish. The dotted circles indicate the boundaries of the retinas. Scale bars: 100 μm. (C) The expression levels of *nrl*, *nr2e3*, and *mafba* in the *nrl*-KO retinas at 2 mpf and 5 mpf were measured via RNA-seq and qPCR. The data are shown as mean with SD (n = 3). *, $p < 0.05$; **, $p < 0.01$.

## Discussion

The zebrafish has become a popular animal model for retinal degeneration and regeneration studies [24–26] and drug screening [51,52]. The developmental models regarding the photoreceptors in zebrafish are mainly based on the assumption that the functions of the key regulatory genes (such as *NRL* and *NR2E3*) are conserved between zebrafish and mammals. However, this assumption has been challenged by several recent studies [29–31]. In this study, we generated an *nrl* knockout zebrafish model and systematically investigated its retinal phenotype. We also characterized the retinal cell populations and the cell-type-specific gene expression profiles in both WT and *nrl* knockout zebrafish through single-cell RNA-seq. The unexpected retinal phenotype and the cellular and molecular findings described here expanded the current developmental model of rod photoreceptors in the zebrafish (Fig 11).

Interestingly, we found that some features of the *nrl* knockout zebrafish, such as the moderate downregulation of rod-specific genes and the presence of rod-cone "hybrid" cells, are in part similar to the phenotype of the *rd7* mice carrying an *Nr2e3* mutation [32]. Meanwhile, the *nr2e3* knockout zebrafish has no rods and shows no expression of rod-specific genes [30], a phenotype similar to that of the *Nrl* knockout mice [10]. Moreover, *Nr2e3* is a direct target gene of *Nrl* in humans and mice [23]. However, knockout of *nrl* does not significantly affect the expression of *nr2e3* in zebrafish (Fig 8C). A previous study has shown that *nr2e3* is expressed prior to *nrl* during the rod genesis in zebrafish [28]. Taken together, we speculate that *nrl* might act in parallel with or even downstream of *nr2e3* in zebrafish. The functions of *NRL* and *NR2E3* and the regulatory relationship between the two genes are not completely conserved between zebrafish and mammals.

The surprising retinal phenotype of the *nrl*-KO zebrafish indeed does not support the use of this zebrafish line as a suitable model of ESCS. However, the reduced number of rods and over-growth of green cones in the *nrl*-KO retinas could still cause a slowly progressing retinal degeneration, as validated by the elevated levels of apoptosis of retinal cells and significant up-regulation of GFAP in Müller glia (Fig 10). The gradual loss of rods results in secondary death of cones in retinitis pigmentosa, the most common type of inherited retinal degenerative disease [48,49]. In addition, our bulk and single-cell RNA-seq data may provide valuable clues for investigating the genes and pathways involved in the occurrence and progression of retinal degeneration. For example, several differentially expressed genes, including *aipl1* (aryl hydrocarbon receptor interacting protein-like 1), *ahrrb* (aryl-hydrocarbon receptor repressor b), *rpgra* (retinitis pigmentosa GTPase regulator a), *nxnl1* (nucleoredoxin like 1), have previously been associated with retinal degenerative diseases [53–57].

One of the major findings in this study is the discovery of two waves of rod genesis occurring at the embryonic and post-embryonic stages through *nrl*-dependent and -independent mechanisms, respectively. The second wave of *nrl*-independent rods may likely represent the rod population constantly generated from the Müller glia-derived unipotent progenitors during the growth of zebrafish [25,28,58], because they show similar developmental stages and retinal locations at the beginning of their genesis. Additionally, we found that the regeneration of rods from the injury-induced Müller glia-derived multipotent progenitors [25] may also be *nrl*-independent (Fig 8). In other words, *nrl* is likely not required for the generation of rods from Müller glial cells.

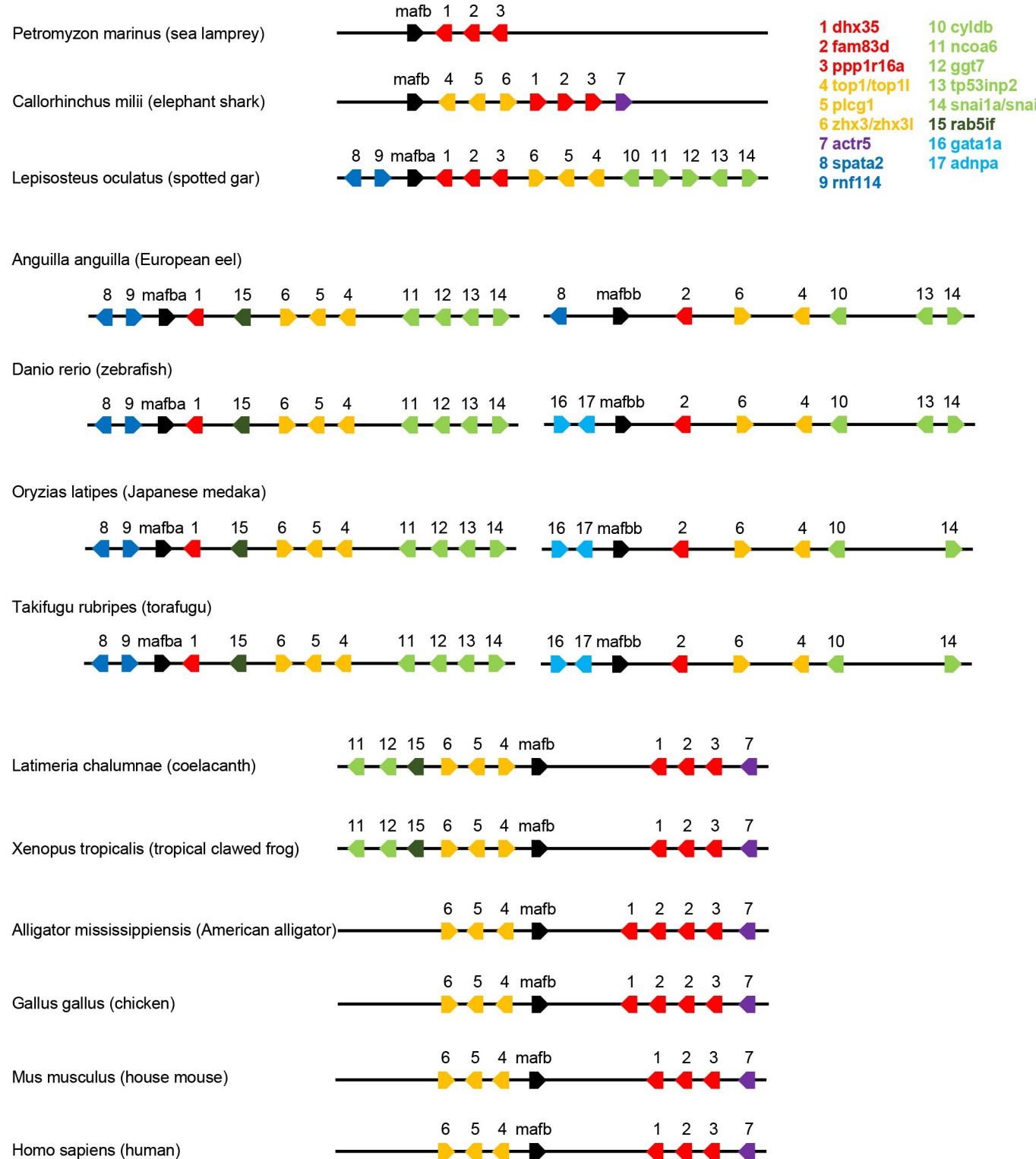

**Fig 9. Gene collinearity of *mafba* among species at different evolutionary positions.** The genes nearby *mafba* and their relative orders were extracted from the genomes of a wide range of vertebrates, including ancient fishes (sea lamprey, elephant shark, spotted gar, and coelacanth), modern fishes (European eel, zebrafish, Japanese medaka, and torafugu), amphibians (tropical clawed frog), reptilians (American alligator), birds (chicken), and mammals (house mouse and human). The *mafba* and *mafbb* genes appear to have originated from the ancestral gene *mafb* through genome duplication specifically in teleost fishes.

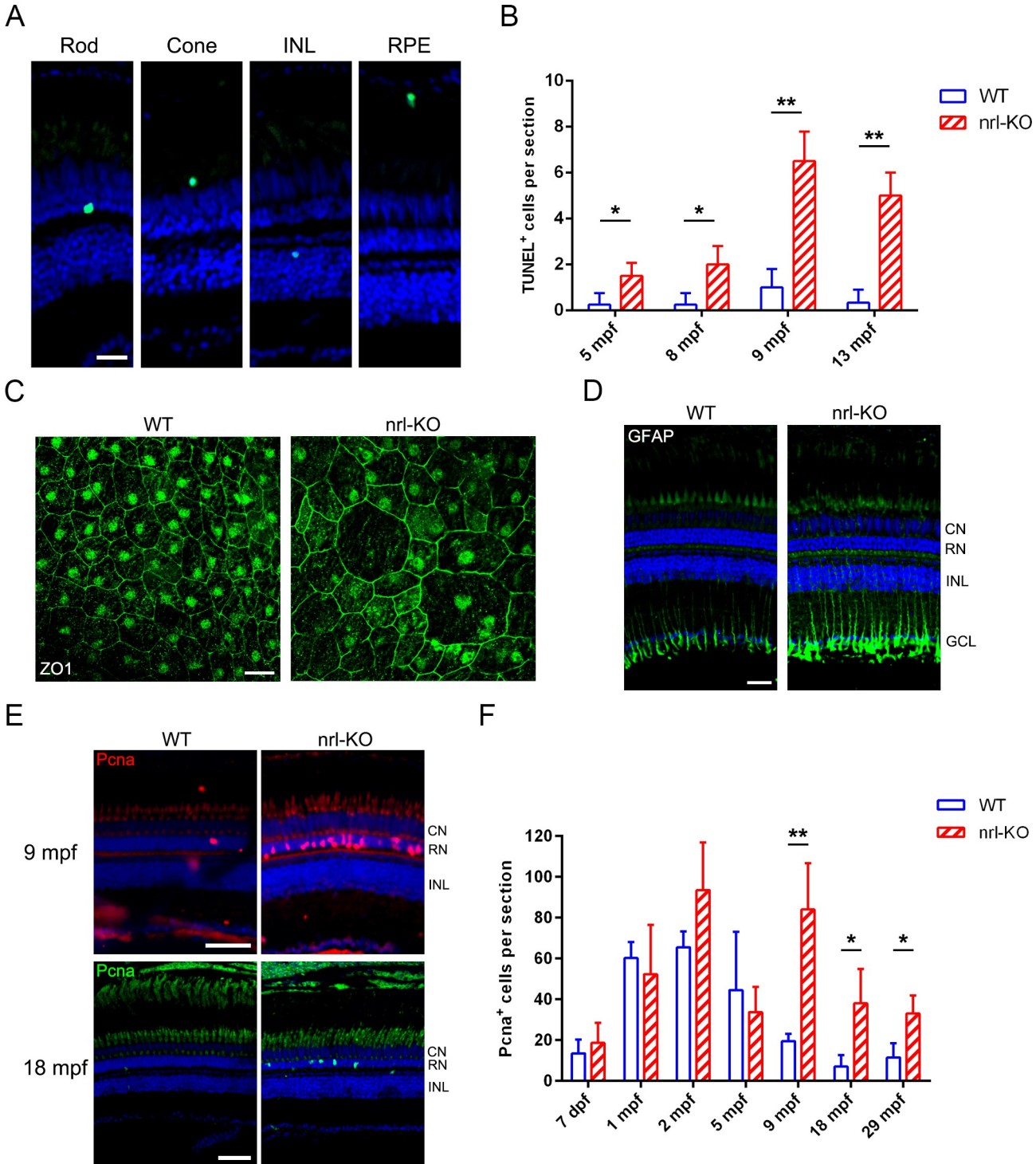

**Fig 10. Progressive degeneration and regeneration in the *nrl*-KO retinas.** (A) The TUNEL assay results revealed that multiple types of retinal cells, including rods, cones, RPE, and inner retinal cells, undergo apoptosis in the *nrl*-KO retinas. Representative images are shown. Scale bar: 20 μm. (B) Quantitation of the apoptotic cells per section from 5 mpf to 13 mpf. The results are shown as mean with SD (n = 3). *, $p < 0.05$; **, $p < 0.01$. (C) RPE morphology in the WT and *nrl*-KO zebrafish at 20 mpf are shown, as assessed by immunostaining the retinal whole-mounts for ZO-1 Scale bar: 20 μm. (D) The up-regulation of GFAP in *nrl*-KO retinas as detected by immunostaining. CN, cone nuclear layer; RN, rod nuclear layer; INL, inner nuclear layer; GCL, ganglion cell layer. Scale bar: 25 μm. (E) Regeneration of photoreceptors was more active in the *nrl*-KO retinas at 9 mpf and 18 mpf, as reflected by the increase in the number of Pcna-positive cells (proliferating cells) in the ONL, compared with the WT levels. Scale bars: 50 μm. (F) Quantitation of the Pcna$^+$ cells located in the ONL per section. The results are shown as mean with SD (n = 4) from 7 dpf to 29 mpf. *, $p < 0.05$; **, $p < 0.01$.

The modified developmental model of rod photoreceptors in zebrafish

**Fig 11. Working model for the development of rod photoreceptors in the zebrafish.** There may be two types of rod precursors defined as *nrl*-dependent and *nrl*-independent in the zebrafish retinas. The former is responsible for the production of rods at embryonic stage, and knocking out *nrl* abolishes the differentiation into rods (left panel). The latter is responsible for the production of rods at the juvenile and adult stages. In the presence of *nrl* and *mafba*, these cells differentiate into rods. Knocking out *nrl* does not severely affect the production of rods from these cells but causes abnormal expression of rod- and cone-specific genes and a progressive increase of green cones (M-cones) with age. Knocking out both *nrl* and *mafba* eliminates all rods, suggesting that *mafba* also plays an important role in the fate determination of this type of rods (right panel).

Interestingly, a recent study has reported the dispensable role of *nrl* in the specification of rods in adult zebrafish but not zebrafish larvae [59], which is partly consistent with our conclusions. Nevertheless, the two studies diverge in several aspects. Firstly, we demonstrated that even in the adult *nrl* knockout zebrafish, the number of rods is reduced to a certain extent, as validated through multiple types of cellular and molecular experiments. Secondly, the study of Oel *et al* has shown an increase of UV cones in the *nrl* mutants at 4 dpf. However, in this study, we found that the mRNA levels of UV opsin (*opn1sw1*) in the *nrl*-KO zebrafish were not up-regulated (Figs 5B and 6C). Lineage tracing experiments have shown that zebrafish rods are unlikely derived from UV- and blue-cone precursors [29]. Besides, our previous study has shown that knocking out *nr2e3* in zebrafish does not increase the number of UV cones, too. Thirdly, we found that the number of green cones is increased in adult *nrl* knockout zebrafish in an age-related manner, and there are rod/green-cone hybrid cells in *nrl*-KO zebrafish. This phenomenon was not mentioned in the study of Oel *et al* [59]. Furthermore, we identified the *mafba* gene as a novel driving gene for the development of rods in zebrafish, which provided a reasonable explanation for the retinal phenotype of the *nrl*-KO zebrafish. Our study focused more on the temporal and spatial changes of retinas in the *nrl* knockout zebrafish at the tissue, cellular and molecular levels, as well as the regulatory mechanisms controlling these biological processes.

The gradual increase in the number of green cones and the presence of rod/green-cone "hybrid" photoreceptors suggest that there may be cell fate transformation between rods and green cones in the *nrl* knockout zebrafish. Among the four types of cone opsins, the green-cone opsin shows the highest similarity and the closest evolutionary relationship to the rod opsin in non-mammalian vertebrates [60]. Thus, we speculated that at least a proportion of rods and green cones may originate from the same precursors in zebrafish, and *nrl* may play a crucial role in the fate choice between rods versus green cones. However, the RH2 (green-cone opsin) and SWS2 opsins are thought to be lost in mammalian ancestors during the adaption of nocturnal lifestyle [61]. This may explain why the S-cones but not the green-cones are alternatively increased in the *Nrl* and *Nr2e3* knockout mice [10,11,32,37].

Through single-cell RNA-seq, we further demonstrated the existence of different rod subtypes and revealed their signature developmental and gene expression patterns (Fig 7 and S1 Table). These findings explain the unique retinal phenotype observed in the *nrl* knockout zebrafish and suggest that zebrafish rods are derived from two different precursors under the control of regulatory genes, such as *nrl* and *mafba*. Recently, scRNA-seq has emerged as a powerful technique to study retinal development and degeneration under normal or pathological conditions [62]. This study also reported the cell compositions and cell-type-specific gene expression profiles in WT and *nrl* knockout retinas. Using these valuable data, we successfully identified the *mafba* gene as a potential driving factor for the formation of the *nrl*-independent rods. This assumption was subsequently validated in the *nrl* and *mafba* double-knockout zebrafish model. Moreover, we identified a cone-specific nuclear receptor gene, *rorcb* (RAR-related orphan receptor C b), to be up-regulated in the *nrl* knockout retinas (Figs 5A and 8A). Whether *rorcb* is responsible for the over-growth of green cones in the *nrl* knockout zebrafish needs to be determined in future studies.

In summary, we comprehensively characterized the retinal phenotype of *nrl* knockout zebrafish at the cellular and molecular levels and discovered novel roles of *nrl* and *mafba* in rod genesis, thereby contributing to the derivation of a more accurate working model for the photoreceptor development in zebrafish. Our findings may promote further studies in the fields of photoreceptor development and evolution, as well as retinal degeneration and regeneration in zebrafish.

## Materials and methods

### Ethics statement

All the procedures involving zebrafish were approved by the Ethics Committee of Huazhong University of Science and Technology.

### Zebrafish lines

WT zebrafish, the *nrl* and *mafba* KO mutants, and the Tg(rho:EGFP) transgenic zebrafish were maintained and bred following the procedure in a previous study [63]. The *nrl* and *mafba* knockout zebrafish were generated using the CRISPR-Cas9 technology as previously described [30,64]. Zebrafish were sacrificed by immersing them in 0.02% MS222 (Sigma, Cat# 886-86-2) solution for 15 minutes until no opercular movement was observed.

### Retinal sectioning and Hematoxylin and Eosin (HE) staining

Zebrafish eyes were isolated and fixed with 4% PFA in PBS at 4˚C overnight. Then, they were incubated in 30% sucrose at 4˚C overnight for cryo-protection. Subsequently, the eyes were embedded in optimal cutting temperature compound (OCT, SAKURA) and sectioned at 10–

15 μm thickness along with the dorsal-ventral orientation through the optic nerve by using a cryostat microtome (CM1860, Leica). The slides were dried at 37˚C for 30 min and then stored at -20˚C. HE staining was performed according to the manufacturer's instructions. The slides were examined and photographed under a light microscope (BX53, Olympus).

## Western blotting, dot bolting and qPCR analyses

Zebrafish eyes were enucleated for the total protein and RNA extractions. For each protein sample, 2–3 eyes from different zebrafish were sonicated in RIPA lysis buffer containing a protease inhibitor cocktail (Sigma, Cat# P2714). The protein samples were mixed with the SDS loading buffer, boiled for 5 min at 95–100˚C, and then stored at -20˚C. To prepare the samples for rhodopsin dot blotting, the zebrafish retinas were directly solubilized in 2% octyl-β-D-glucopyranoside (Aladdin, Shanghai, China) in PBS with sonication. The supernatant containing rhodopsin was collected by centrifugation (4˚C, 20000 g, 30 min). For each RNA sample, three eyes (lens removed) from different zebrafish were homogenized in the RNAiso Plus reagent (Takara, Cat# 9108). Total RNA was extracted following the manufacturer's instructions and then stored at -70˚C. Western blot and qPCR were performed as described previously [63]. The antibodies and primers used are listed in S2 and S3 Tables, respectively.

## Immunofluorescence staining and TUNEL assay

Immunofluorescence staining on retinal sections was performed as described previously [63]. For retinal whole-mount preparation, zebrafish eyes were fixed in 4% PFA in PBS at room temperature for 30 min and dissected under a stereomicroscope. The anterior segment, sclera, and choroid were removed, and the retinas were cut into several pieces. The whole-mount retinas were re-fixed in 4% PFA for 30 min, washed three times with PBS, and then blocked and permeabilized in PBST (0.5% Triton X-100) with 10% goat serum overnight at 4˚C. Afterward, the retinas were incubated with the primary antibody solution overnight at 4˚C, washed three times with PBST for 30 min each, incubated with the secondary antibody solution for 4 h at room temperature, and washed again three times as described above. To visualize the entire retina, each region within the same retina was photographed, and the resulting series of photographs were combined into a panoramic image. TUNEL (terminal deoxynucleotidyl transferase biotin-dUTP nick end labeling) assay was performed on retinal sections following the manufacturer's instructions (Cat # 11684795910, Roche).

## Ultrastructural analysis of the rod outer segments

Transmission electron microscopy (TEM) was performed as described previously [63]. Briefly, zebrafish eyes were fixed in the fixative solution (2.5% glutaraldehyde, 0.1 M PBS buffer, pH 7.0) at room temperature for 30 min, dissected into proper size, and then continued to be fixed overnight at 4˚C. After three washes with PBS, the eyes were further fixed in 1% osmium tetroxide for 2 h at room temperature, and then dehydrated through consecutive incubations (15 min each) in 50, 70, 80, 90, 95, and 100% ethanol. Subsequently, the eyes were incubated in acetone for 20 min, consecutively treated with 50% (1 h), 75% (3 h) and 100% (overnight) epoxy resin (mixed with acetone, v/v), and then heated at 70˚C overnight. The resulting embedded eyes were sliced into ultrathin sections (70 nm) by using a Reichert-Jung ultramicrotome (Leica). The sections were stained with 3% uranyl acetate and 3% lead citrate for 15 min. The dorsal region of the retina was examined via a transmission electron microscope system (HT7700, Hitachi).

### Tracing the fluorescence of rods in the Tg(rho:EGFP) transgenic zebrafish

At 3–17 dpf, the zebrafish were sacrificed and fixed with 4% PFA in PBS for 15 min at room temperature. After washed three times with PBS, the eyeballs were isolated, and the sclera/choroid was removed under a stereomicroscope. The eyeballs were depigmented in a solution containing 3% $H_2O_2$ and 1% KOH at room temperature for 5–10 min, washed three times with PBS for 10 min each, mounted in 0.8% low-melting agarose, and then immediately photographed under a fluorescence microscope (Eclipse 80i, Nikon). To visualize the fluorescence of the whole retina comprehensively, two photographs were taken at different Z-positions and then merged into one image for each eyeball.

At 1–3 mpf, the zebrafish were sacrificed and the eyeballs were isolated and fixed with 4% PFA for 30 min at room temperature. Afterward, the eyeballs were washed three times with PBS, and then the anterior segment, sclera, and choroid were removed under a stereomicroscope. The retinas were depigmented as described above. Subsequently, the retinas were cut into several pieces and mounted in PBS on microslides. The samples were immediately photographed under a fluorescence microscope (Eclipse 80i, Nikon). A series of local images from a single retina were merged to visualize the whole structure.

### *In situ* hybridization on retinal sections

A cDNA fragment of the *hmgn2* gene was amplified from the retina cDNA library by using the primers (F: CGCGAGGTTGTCTGCTAAAC, R: GGTGAAAACCCTTCCGAAAACA) and validated via Sanger sequencing. The T7 promoter was added to the 5' and 3' ends to synthesize the sense and anti-sense probes by using the DIG RNA Labeling Kit (Roche, Cat# 11175025910). *In situ* hybridization was performed as previously described [65,66] with minor modifications. The signals were detected and developed using an alkaline-phosphatase-conjugated anti-DIG antibody (Roche, Cat# 11093274910) and the NBT/BCIP solution (Roche, Cat# 11681451001) following the manufacturer's instructions. The slides were examined and photographed under a light microscope (BX53, Olympus).

### RNA-seq and bioinformatic analysis

The total RNA samples were quantified and qualified using an Agilent 2100 Bioanalyzer (Agilent Technologies) and NanoDrop (Thermo Fisher Scientific). RNA samples with RIN $\geq$9.5 and A260/280 $\geq$1.9 were used for RNA-seq. The next-generation sequencing and data analysis were performed by GENEWIZ (Suzhou, China). Briefly, the NEBNext Ultra RNA Library Prep Kit for Illumina was used for library preparations. Next-generation sequencing was carried out on an Illumina HiSeq instrument using the 2 x 150 bp paired-end configuration according to the manufacturer's protocol.

High-quality clean data were generated using Trimmomatic (v0.30) and aligned to the zebrafish reference genome (GRCz11 from Ensembl) via the Hisat2 (v2.0.1) software. Gene expression levels were estimated via HTSeq (v0.6.1). Differential gene expression analysis was performed using the DESeq Bioconductor package. After adjustment using the Benjamini and Hochberg's approach for controlling the false discovery rate, the *p*-value threshold was set to <0.05 to detect differentially expressed genes. GO and KEGG enrichment analyses were then performed to enrich the differential expression genes in GO terms and KEGG pathways.

### Preparation of the retinal single-cell suspensions

The eyes of 5-month-old WT and *nrl*-KO zebrafish were isolated and dissected in cold HBSS buffer by using microsurgical scissors and forceps under a stereomicroscope. The

anterior segment, sclera, and choroid were removed. The retina was transferred to a well of a 12-well plate with 1 ml of the activated papain dissociation solution (20 U/ml papain, 1 mM L-cysteine, 0.5 mM EDTA in HBSS, pH 6.0–7.0; activated via incubation at room temperature for 30 min), and incubated at 28°C for 10–15 min with gentle agitation. The retina was aspirated and expelled through a wide-orifice pipette tip once every 2 min to promote the dissociation and prevent the aggregation of retinal cells. Any visible piece of non-dissociated tissue was removed using fine forceps. Subsequently, the cell suspension was filtrated through a 40 μm cell strainer (Corning, Cat# 431750), and then pelleted through centrifugation at 300 g for 5 min at room temperature. The cells were washed twice with HBSS buffer (without $Ca^{2+}$ or $Mg^{2+}$) containing 0.04% BSA and then resuspended in the same solution. The cell viability was determined via trypan-blue staining and ensured to be over 80%. The cell concentration was adjusted to ~1000 cells/μl and placed on ice (<30 min).

## Single-cell RNA-seq (scRNA-seq) and bioinformatic analysis

Retinal single-cell suspensions were loaded onto the 10x Genomics Chromium Single Cell system using the 3' Reagent Kits (v3 chemistry) at Genedenovo Biological Technology Co., Ltd. (Guangzhou, China). Approximately 10,000 live cells were loaded per sample. Single-cell GEM generation, barcoding, and library preparation were performed according to the user guide. High-throughput sequencing was performed on the HiSeq 2500 systems (Illumina) at Genedenovo. The Cell Ranger pipelines were used to align the reads to the reference genome (GRCz11 from Ensembl), generate feature-barcode matrices, and perform clustering and gene expression analysis.

Seurat [67] was used for quality control, gene expression normalization, principal component analysis (PCA), and cell clustering. In brief, cells with a clear outlier number of genes (potential multiplets) and a high percentage of mitochondrial genes (potential dead cells) were excluded using the "FilterCells" function. The global-scaling normalization method "LogNormalize" was employed to normalize the gene expression measurements for each cell. PCA was performed to reduce the dimensionality of the dataset. Seurat was used for clustering cells based on their PCA scores. To determine the most contributing PCs, a resampling test inspired by the jackStraw procedure was implemented. The significant PCs with a strong enrichment of low $p$-value genes were used to identify the unsupervised cell clusters. t-SNE (t-distributed Stochastic Neighbor Embedding) was used to visualize and explore the distribution of the cells in a low-dimensional space (2D-plot). For every single cluster, up-regulated genes were identified via the likelihood-ratio test using the "FindAllMarkers" function in Seurat package. The top 20 up-regulated genes were selected as marker genes for each cluster. GO and KEGG enrichment analyses were performed to identify the main biological functions and the significantly enriched metabolic or signal transduction pathways among the differentially expressed genes.

## Statistical analysis

The unpaired Student's two-tailed t-test was performed using the GraphPad Prism 6 software to determine the statistical significance of the difference between two groups. A $p$-value less than 0.05 was considered to indicate statistical significance (*, $p < 0.05$; **, $p < 0.01$; ***, $p < 0.001$). All the data are presented as mean with SD. All the experiments were independently repeated at least three times. The numerical data underlying the graphs or summary statistics in this study is included in S1 Data.

## Supporting information

**S1 Fig. Histological analysis of the WT and *nrl*-KO zebrafish retinas.** (A and D) HE staining of the retinal sections from the dark- and light-adapted WT and *nrl*-KO zebrafish at 1 mpf and 2 mpf, respectively. Scale bars: 50 μm in (A) and 100 μm in (D). (B and E) Enlarged images of the retinal regions indicated by red arrows in (A) and (D). The green vertical lines indicate the thicknesses of the PR+RPE layer in (B) and the OS layer in (E). Scale bars: 50 μm in (B) and 25 μm in (E). PR, photoreceptor; RPE, retinal pigment epithelium; OS, outer segment; IS, inner segment; ONL, outer nuclear layer; INL, inner nuclear layer; GCL, ganglion cell layer. (C and F) Quantitation the thicknesses of the PR+RPE layer in (B) and the OS layer in (D), as measured every 100 μm from the optic nerve to the edges of the ventral and dorsal retinas. The results are shown as mean with SD (n = 6 and n = 3, respectively). Blue lines, WT group; red lines, *nrl*-KO group.
(TIF)

**S2 Fig. Overall views of the rod outer segments in the WT and *nrl*-KO retinas.** The rod outer segments were labeled with the anti-Rho antibody on retinal sections from the WT and *nrl*-KO zebrafish at 14 dpf (A), 1 mpf (B), 2 mpf (C), and 4 mpf (D). The regions nearby the ciliary marginal zone (labeled with boxes) showed no or weak fluorescence signal of rods in the *nrl*-KO retinas at 14 dpf and 1 mpf. Scale bars: 50 μm.
(TIF)

**S3 Fig. Quantitation of the thickness of the rod OS layer and the density of the rod OSs in WT and *nrl*-KO zebrafish.** (A) The rod outer segments were visualized via immunostaining on retinal sections (see Figs 1D and S2) from 1 mpf to 18 mpf using the anti-Rho antibody. The thickness of the rod OS layer was measured in the dorsal-middle regions of the WT and *nrl*-KO retinas. The results are shown as mean with SD (n = 6). **, $p < 0.01$. (B) Immunostaining was performed on flattened whole-mount retinas to visualize rod outer segments at 2 mpf (see Fig 2). The density of the rod OSs (the number in a 100 μm x 100 μm region) were measured in the dorsal regions of the WT and *nrl*-KO retinas. The results are shown as mean with SD (n = 6). **, $p < 0.01$.
(TIF)

**S4 Fig. Decreased expression of Rho in the *nrl*-KO retinas as detected by dot blotting.** A series of gradient dilutions of retinal protein extracts from the WT and *nrl*-KO zebrafish at 9 mpf were spotted onto the NC membranes and immunoblotted with the anti-Rho antibody. Tubulin served as a loading control. By comparing the 10x dot in the *nrl*-KO group with the 20x and 50x dots in the WT group, Rho was estimated to be 2.5–5 fold downregulated.
(TIF)

**S5 Fig. Ultrastructure of the rod outer segments in the WT and nrl-KO retinas at 9 mpf.** (A) The outer segments of rods are shown in low-magnification TEM image. BM, Bruch's membrane. RPE, retinal pigment epithelium. Scale bar, 10 μm. (B) Single outer segments of rods are shown in high-magnification TEM images. Compared with the rods in the WT group, most of the rods in the nrl-KO retinas showed a relatively loose membrane disc structure. There was no large difference in the length of outer segments between the WT and nrl-KO rods. OS, outer segment. IS, inner segment. Scale bar, 2 μm.
(TIF)

**S6 Fig. Cell-type specific expression profiles of the marker genes for each retinal cell type.** All of the unsupervised cell clusters identified by scRNA-seq were matched to the known retinal cell types.
(TIF)

**S7 Fig. Up-regulation of *hmgn2* specifically in the rods of *nrl*-KO zebrafish.** (A) *In situ* hybridization analysis of *hmgn2* on retinal sections from the WT and *nrl*-KO zebrafish at 3 mpf. Scale bars: 100 μm. (B) Enlarged images of the dorsal and ventral regions of WT and *nrl*-KO retinas are shown. Scale bars: 25 μm.
(TIF)

**S8 Fig. Generation of the *mafba* knockout zebrafish via the CRISPR-Cas9 technology.** (A) The mean expression levels of *nrl*, *nr2e3*, *mafba*, and other large MAF family genes in each retinal cell type of the zebrafish are shown. Like nrl and nr2e3, the *mafba* gene (indicated by green bars) was highly expressed in all the three types of rods like *nrl* and *nr2e3*. Additionally, *mafba* was also expressed in the microglia. (B) The protein domains, gene structure, and CRISPR-Cas9 target site of *mafba* are shown. (C) Validation of the *mafba* knockout zebrafish carrying the homozygous del7 mutation (c.175_181del7, p.T59Sfs*43) via sequencing. The red box (upper panel) and the red line (lower panel) indicate the deleted 7 bp region in the *mafba* gene.
(TIF)

**S1 Text. The legends of the Supplementary figures.**
(DOC)

**S1 Table. The top 20 marker genes for the three rod clusters identified via scRNA-seq.**
(DOC)

**S2 Table. The antibodies used in this study.**
(DOC)

**S3 Table. The primers used for qPCR in this study.**
(DOC)

**S1 Dataset. Differentially expressed genes between WT and *nrl*-KO retinas identified via RNA-seq.**
(XLSX)

**S2 Dataset. Functional categories of the top 100 differentially expressed genes between WT and *nrl*-KO retinas.**
(XLSX)

**S3 Dataset. Retinal cell clusters and the corresponding gene expression data from the scRNA-seq results.**
(XLSX)

**S4 Dataset. Marker genes for each cell cluster identified via scRNA-seq.**
(XLSX)

**S1 Data. The numerical data underlying the graphs or summary statistics in this study.**
(XLSX)

# Acknowledgments

We thank Dr. Jian Zou (Eye Center of the Second Affiliated Hospital School of Medicine, Institutes of Translational Medicine, Zhejiang University) for providing the Tg(rho:EGFP) zebrafish line. We thank Yuan Xiao and Zhenfei Xing for providing the TEM service (Analysis and Testing Center, Institute of Hydrobiology, CAS).

## Author Contributions

**Conceptualization:** Fei Liu, Chengqi Xu, Daji Luo, Mugen Liu.

**Data curation:** Fei Liu.

**Formal analysis:** Fei Liu, Yayun Qin.

**Funding acquisition:** Fei Liu, Shanshan Yu, Zhaohui Tang, Mugen Liu.

**Investigation:** Fei Liu, Yayun Qin, Yuwen Huang, Pan Gao.

**Methodology:** Fei Liu, Yayun Qin, Yuwen Huang, Pan Gao, Jingzhen Li.

**Project administration:** Fei Liu, Zhaohui Tang, Mugen Liu.

**Resources:** Danna Jia, Xiang Chen, Yuexia Lv, Jiayi Tu, Kui Sun, Yunqiao Han.

**Supervision:** Fei Liu, Daji Luo.

**Visualization:** Fei Liu, Yayun Qin.

**Writing – original draft:** Fei Liu, Yayun Qin.

**Writing – review & editing:** Fei Liu, James Reilly, Xinhua Shu, Qunwei Lu, Chengqi Xu, Daji Luo, Mugen Liu.

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
