## [Decision Letter · Decision Letter 0]

7 Nov 2021

Dear Dr Liu,

Thank you very much for submitting your Research Article entitled 'Rod-genesis driven by mafba in an nrl knockout zebrafish model with altered photoreceptor compositions and progressive retinal degeneration' to PLOS Genetics.

The manuscript was fully evaluated at the editorial level and by independent peer reviewers. The reviewers appreciated the attention to an important problem, but raised some substantial concerns about the current manuscript. Based on the reviews, we will not be able to accept this version of the manuscript, but we would be willing to review a much-revised version. We cannot, of course, promise publication at that time.

If you decide to revise the manuscript for further consideration at PLOS Genetics, please aim to resubmit within the next 60 days, unless it will take extra time to address the concerns of the reviewers, in which case we would appreciate an expected resubmission date by email to plosgenetics@plos.org.

[LINK]

We are sorry that we cannot be more positive about your manuscript at this stage. Please do not hesitate to contact us if you have any concerns or questions.

Yours sincerely,

Anand Swaroop

Guest Editor

PLOS Genetics

Gregory Barsh

Editor-in-Chief

PLOS Genetics

This manuscript presents interesting findings that complement and expand on another recently published manuscript by Oel et al. The two reviewers have raised important issues. In general, the manuscript has numerous grammatical and spelling mistakes, and many sentences require revision. A better reorganization of the manuscript may allow the authors to focus on key points. In addition, the authors should discuss their findings by incorporating the work from Oel et al and describing distinctions from mammals.

Reviewer's Responses to Questions

**Comments to the Authors:**

Reviewer #1: Review uploaded

Reviewer #2: PGENETICS-D-21-01260 Liu et. al “Rod-genesis driven by mafba in an nrl knockout zebrafish model with altered photoreceptor compositions and progressive retinal degeneration”

This manuscript describes the identification of a gene, mafba, as a novel regulator for nrl-independent rod formation in zebrafish. The authors generated an nrl knockout zebrafish using CRISPR-Cas9, and saw a reduction but not total elimination of rods along with an overgrowth of green cones. There are major and minor comments to be addressed prior to publication. It is strongly recommended that the manuscript be restructured such the main thrust of the data are in the main body of the text and the supplemental data for incremental data for ease of reading.

Major comments:

Abstract/Author Summary

The authors state that “abnormal gene expression caused progressive retinal degeneration and subsequent regeneration in nrl knockout zebrafish.” Since zebrafish are able to regenerate damaged retinas through the Müller glia-derived progenitors (PMCID: PMC2948409), could these be the source of the second wave of rod genesis they mention earlier in the abstract? This is not addressed in the text. Since this is a viable option for data seen, this needs to be addressed and considered.

The authors state that the importance of these findings in zebrafish “highlight the conserved and species-specific regulatory mechanisms of photoreceptor development and maintenance” however the difference between zebrafish photoreceptor development and mammalian photoreceptor development implies they are not a great model for human disease mechanism studies. This is especially true since these results are in contrast with the Nrl knockout mice. From this point, it would be reasonable if the authors restructure their paper such that they are making the point that zebrafish are not a good model for human disease, and they have discovered data underpinning zebrafish development which may not be applicable to human retinal development. This reviewer realizes this is not an exciting or is perhaps controversial, but the data presented here do not correlate with mouse development, and therefore may be different in humans as well. Either way, they do not support the statement above.

“NRL is the most important gene for the occurrence and function of rod cells in mice and humans” – This sort of declarative statement can be refuted and as such care should be taken in overstatements like this throughout the text.

Line 36: “over-grown of” should be replaced with “over-growth of” or “compensation by”

Line 60-61: “breaking the function”? this phrase is confusing

Results

Figure 1

Zebrafish Nrl antibodies couldn’t be used for western or IF. This in concert with an increase in mRNA does not result in confidence that they are getting a true knockout of nrl.

Figure S1

Why are they comparing different ages 1 mpf and 2 mpf WT and KO, respectively?

They have rod nuclei (presumably) in H&E stained sections at 2 mpf in KO but decrease in thickness of PR+RPE layer – is this due to loss of outer segments?

Figure 2/S2

The authors state that thickness of rod OS and number of rod nuclei is significantly reduced – but this is not quantified anywhere in figure2 or S2. Likewise, it is stated on line 163, “we didn’t observe the fluorescence of rods until 7 dpf in KP zebrafish”. Please see figure and legend S4A/B. There are some rods; little, but some.

What is the difference in Rho/anti-Rho labeling? Why do they go back and forth between red and green? This is confusing.

Figures 3 & S3

They state that density of rod OS significantly reduce in nrl-KO zebrafish and show staining in S3 but it is not quantified anywhere in S3.

Same with the number of rod numbers “obviously reduce” on page 10. They never quantify.

In the text, they talk about mRNA levels of cone specific genes being significantly changed and reference 3C, but this is not what 3C shows. I assume they mean 3B.

Figures 4 & S4

A – it is unclear if these are separate animals/in triplicate or duplicate/ or if they’re different time points quantified in B

B – It is difficult to tell which bars go with what WT in the legend. Consider separating by age and comparing WT with KO. S4B should be in the manuscript, not supplemental.

Additionally, the authors state "the fluorescence of every single rod was also significantly lower than WT zebrafish." Is this quantified? It needs to be in order to state this. WT is oversaturated in the image; what do the data look like in sub-saturating conditions?

Line 174/5 states that rods were absent in marginal zones, yet this reviewer sees rods. Indeed, the authors say in the figure legend for S4 that "arrows indicate retinal regions without rods." I don't see that; when I increase the size of the figure in C, I do see what appear to be rods.

The authors state that there is limited distribution and decreased expression of rod opsin on retinal sections of KO at 1 mpg. Is this due to transgene? Transmission electron microscopy should be performed to determine the ultrastructure of the OS in the KO animals. In this case, the OS should be shorter and thinner with a decrease in rhodopsin expression. Additionally, dot blots are warranted to quantitatively show a decrease in rhodopsin expression.

Figure 5

The authors show that mafba mRNA levels are decreased at 2, and 5 months in the nrl KO. If it is compensatory or involved in the “nrl-independent” pathway, why would the mRNA levels be lower?

Figure 6

The authors make the comment that mafba + nrl double knockout eliminates rods (nrl KO alone does not and mafba KO alone dies at 9 dpf). Since it is a global KO of mafba and nrl, could it be that they are affecting the ability of new progenitors being formed from Müller glia etc. Double knockout with a gene that is lethal at 9 dpf could have many off target issues.

**Have all data underlying the figures and results presented in the manuscript been provided?**

Reviewer #1: Yes

Reviewer #2: Yes

PLOS authors have the option to publish the peer review history of their article (what does this mean?). If published, this will include your full peer review and any attached files.

Reviewer #1: No

Reviewer #2: No

---

## [Decision Letter · Decision Letter 1]

1 Feb 2022

Dear Dr Liu,

Thank you very much for submitting your Research Article entitled 'Rod-genesis driven by mafba in an nrl knockout zebrafish model with altered photoreceptor compositions and progressive retinal degeneration' to PLOS Genetics.

The manuscript was fully evaluated at the editorial level and by independent peer reviewers. The reviewers appreciated the attention to an important topic but identified minor issues that we ask you address in a revised manuscript

We therefore ask you to modify the manuscript according to the review recommendations. Your revisions should address the specific points made by each reviewer.

[LINK]

Yours sincerely,

Anand Swaroop

Guest Editor

PLOS Genetics

Gregory Barsh

Editor-in-Chief

PLOS Genetics

As indicated by both reviewers, this is a much improved manuscript. Overall, this is an excellent manuscript and the results go farther than what has been shown by Oel et al. In fact, the data are consistent with Kim et al. Dev Cell 2016 suggesting at least two different types of rods. It appears that mafba and nrl have complementary and probably synergistic functions in zebrafish and that NRL became the primary rod determinant in mammals. Changes in regulation of gene expression of these two transcription factors can account for evolutionary differences (evo-devo). Perhaps the authors can discuss some of these aspects in Discussion

Reviewer's Responses to Questions

**Comments to the Authors:**

Reviewer #1: The authors have addressed my major concerns in this revised manuscript. There are two points that I believe should be addressed. First, the authors have expanded their comparison of the previous (overlapping work) of Oel et al. on p. 21-22. However, I feel the text included in the response to the reviewers is more direct and compelling than that in the paper itself. In fact, I suggest the authors incorporate the response directly into the paper. I also think a few sentences in the introduction to signal that there has been similar work published would help the reader and put both studies in perspective. But this may be a matter of taste. Second, the writing has improved, but it still suffers from significant language problems. I recommend that the paper be edited once more, I believe it will dramatically improve the impact.

Reviewer #2: PGENETICS-D-21-01260

This is a much-improved revision of a manuscript describing the identification of mafba as a novel regulator for nrl-independent rod formation in zebrafish. The authors addressed the major and minor concerns posed by the two reviewers. With minor edits, this reviewer agrees this paper should be published in PLOS Genetics.

Minor edits include:

• Lines 181-182: Most likely, the separation between the disks is due to artifacts of transmission electron microscopy. The last sentence implies nrl is directly involved in disk integrity during outer segment turnover. To state this, even with the qualifier “seemingly”, one would need to do cryoelectron tomography to unveil the disk ultrastructure with that level of precision. To avoid having to do this, I would consider deleting this sentence.

• Line 396: change “What’ more important” with “Importantly”

• Line 399: change “tissular” with “tissue”

• Line 432: this sentence should end with “… and retinal degeneration and regeneration in zebrafish.”

**Have all data underlying the figures and results presented in the manuscript been provided?**

Reviewer #1: Yes

Reviewer #2: Yes

PLOS authors have the option to publish the peer review history of their article (what does this mean?). If published, this will include your full peer review and any attached files.

Reviewer #1: No

Reviewer #2: No

---

## [Editor Report · Decision Letter 2]

17 Feb 2022

Dear Dr Liu,

We are pleased to inform you that your manuscript entitled "Rod genesis driven by mafba in an nrl knockout zebrafish model with altered photoreceptor composition and progressive retinal degeneration" has been editorially accepted for publication in PLOS Genetics. Congratulations!

Yours sincerely,

Anand Swaroop

Guest Editor

PLOS Genetics

Gregory Barsh

Editor-in-Chief

PLOS Genetics

Comments from the reviewers (if applicable):

**Data Deposition**

http://datadryad.org/submit?journalID=pgenetics&manu=PGENETICS-D-21-01260R2

**Press Queries**

---

## [Editor Report · Acceptance letter]

25 Feb 2022

PGENETICS-D-21-01260R2 

Rod genesis driven by mafba in an nrl knockout zebrafish model with altered photoreceptor composition and progressive retinal degeneration 

Dear Dr Liu, 

We are pleased to inform you that your manuscript entitled "Rod genesis driven by mafba in an nrl knockout zebrafish model with altered photoreceptor composition and progressive retinal degeneration" has been formally accepted for publication in PLOS Genetics! Your manuscript is now with our production department and you will be notified of the publication date in due course.

With kind regards,

Zsofia Freund

PLOS Genetics

On behalf of:
